# A large C-terminal Rad52 segment acts as a chaperone to Form and Stabilize Rad51 Filaments

Emilie Ma [1,2], Fadma Lakhal[3], Eleni Litsardaki [4,5], Myriam Ruault [3], Maxime Audin [4,5], Natacha Levrier [1,2,4,5], Emilie Navarro [1,2], Mickaël Garnier [3], Laurent Maloisel[1,2], Jordane Depagne[6,7], Clémentine Brocas [6,7], Aurelien Thureau [8], Didier Busso [6,7], Xavier Veaute [6,7], Raphaël Guerois [4,5], Angela Taddei [3,9] ✉, Françoise Ochsenbein [4,5,9] ✉ & Eric Coïc [1,2,9] ✉

Homologous recombination (HR) is essential for the repair of DNA double-strand breaks and the restart of stalled replication forks. A critical step in HR is the formation of Rad51 nucleofilaments, which perform homology search and strand invasion of a homologous DNA sequence required for repair synthesis. In the yeast *Saccharomyces cerevisiae*, Rad52 facilitates Rad51 nucleofilament formation by mediating Rad51 loading onto ssDNA and counteracting Rad51 filament dissociation by the DNA translocase Srs2. The molecular basis of these two Rad52 functions remains unclear. Our integrative structural analyses of the Rad51-Rad52 interaction, combining NMR, SAXS, and modeling, reveal that an 85-residue segment of Rad52, conserved in fungi, folds upon binding to a broad surface of a Rad51 monomer. Notably, it includes an FxxA motif conserved in the BRC repeats of BRCA2 and at the Rad51-Rad51 interface. This binding mode was validated through an extensive set of mutations. Using in vivo assays and a functional fluorescent GFP-Rad51 fusion protein, we demonstrated that this entire segment is critical for Rad51 filament formation. These findings highlight how Rad52 functions as an assembly chaperone by preventing Rad51 oligomerization, promoting nucleation of Rad51 nucleofilaments on ssDNA, and counteracting the effects of Srs2 on destabilizing Rad51 filaments.

Homologous recombination (HR) is an important pathway for DNA double-strand break (DSB) repair that exists in all kingdoms of life[1]. HR also contributes to the robustness of DNA replication through multiple mechanisms, including stabilizing and protecting replication forks[2,3]. However, HR is a double-edged sword: it is essential for maintaining genome stability and generating genetic diversity, but it can also lead to genome instability and cell death[4,5]. In particular, excessive or unscheduled HR can generate highly toxic, unprocessed HR intermediates[6–8]. In eukaryotes, the recombinase Rad51 plays a central role in HR by forming nucleoprotein filaments on single-stranded DNA (ssDNA) generated by nucleases from DSBs or from stalled DNA replication forks[9–15]. These filaments scan the genome for homologous DNA and then perform synapsis and strand invasion, enabling faithful DNA synthesis and repair[16–20]. Although Rad51 filament formation has been extensively studied, it is still unclear how the regulation of its formation/stability ensures safe HR. Positive

regulators of Rad51 filaments support their formation and stability, whereas negative regulators favor their dissociation, limit their extension, and suppress their potential toxicity[21].

A key activity of positive regulators of Rad51 filaments is to displace RPA (Replication Protein A), which binds to ssDNA at the beginning of HR, to facilitate Rad51 filaments formation (also referenced as the mediator function of positive regulators). In yeast, this activity is mainly carried out by the Rad52 protein, whereas in mammals it is performed by BRCA2[22–30]. Notably, yeast Rad52 includes an annealase domain that, while absent in BRCA2 is conserved in mammalian RAD52[31,32]. In yeast and mammals, Rad51 paralogs are also involved in this process[33–40]. Rad52 was primarily viewed as a mediator promoting the replacement of RPA by Rad51. There is now increasing evidence for additional roles of Rad52 in Rad51 filament formation and stability. In particular, we have gathered evidence suggesting that Rad52 might also protect Rad51 filaments from dissociation by the Srs2 DNA translocase[41].

In *Saccharomyces cerevisiae*, Rad52 plays one of the most prominent roles in HR, as evidenced by the severe phenotype caused by *RAD52* null mutations: γ-ray sensitivity and severely reduced levels of both mitotic and meiotic HR[42]. In addition, Rad52 facilitates tolerance to DNA damage not only through HR but also through the translesion polymerase pathway[43]. It also plays a role in the assembly and maintenance of spindle assembly checkpoint activity during mitosis[44]. Rad52 also promotes single-stranded DNA annealing[31,32], an activity required in *S. Cerevisiae* for the post-invasion steps of DSB repair[45,46] as well as the non-conservative mechanism of single strand annealing (SSA)[47].

Yeast Rad52 consists of two domains: the well folded N-terminal domain (1-205), which is similar to human RAD52 and involved in DNA binding, and the less conserved C-terminal region (206-471) (Fig. 1A). Yeast and human RAD52 N-terminal domain form a ring-shaped oligomeric structure around which ssDNA can wrap[25,48] (Supplementary Fig. 1A). This domain catalyzes ssDNA annealing as an oligomeric form in vitro[31]. Although the existence of such oligomers in vivo has not yet been demonstrated, the estimated diffusion coefficients of Rad52 in living cells are consistent with the values predicted for monomeric and multimeric forms of Rad52[49]. Rad52 C-terminal domain contains non-overlapping RPA and Rad51 interaction sites thought to stimulate RPA displacement from DNA and to recruit Rad51 to DNA. Two conserved Rad51 binding motifs, FVTA and YEKF, have been previously described, using a residue numbering that starts 33 amino acids downstream of the first start codon[50–52]. However, the molecular basis for this activity is not well understood. Furthermore, Rad51 was shown to self-associate in solution, which may be a barrier to its loading on DNA. The Rad52 C-terminus was reported to disrupt Rad51 oligomers and form a heterodimeric complex with Rad51, which may be an important feature of Rad52 mediator activity[52]. The Rad51 polymerization motif FVTA is conserved in Rad52 C-terminus and essential to disrupt Rad51 oligomers. A similar domain FxxA is found in the BRC repeats of BRCA2 that also disrupts Rad51 oligomer[23,53–56]. More recently, an interaction between the N- and C-terminal domains was proposed to be important for Rad51 filament formation[25].

Previously, we showed that while Rad52 is essential for Rad51 filament formation, its interaction with Rad51 may not be. We found that Rad52 mutants affecting Rad51 interaction are still proficient in Rad51 filament formation, but the filaments formed are more sensitive to Srs2 helicase activity[41,57]. Furthermore, in Srs2-deficient cells with strong native Rad52-Rad51 interactions, we observed the accumulation of aberrant Rad51 filaments leading to severe toxicity, probably due to obstructions in DNA metabolisms[8]. Therefore, our current model states that Rad51 filament homeostasis is finely regulated by the interplay between Rad52 and Srs2.

To better understand the molecular basis of the interplay between Rad52 and Rad51, we performed integrative structural analyses combining NMR, SAXS, and modeling. We then designed interaction mutants and assessed their defects using in vivo assays and a functional fluorescent GFP-Rad51 fusion protein[19]. Our results reveal that an 85-residue segment conserved in fungi folds upon binding to a broad surface of Rad51. This interaction covers the dimerization interface of Rad51 within the filament and prevents Rad51 oligomerization in vitro. We propose that this segment of Rad52 functions as an assembly chaperone, preventing Rad51 oligomerization, promoting the formation of a functional nucleofilament, and counteracting the destabilization action of Srs2.

## Results

### A large segment of 85 residues in the center of the fully disordered Rad52 C-terminal domain interacts with Rad51

Although the C-terminal region of Rad52 (206–471, Fig. 1A, note that the Rad52 amino acids are numbered from the first AUG codon in *RAD52* mRNA[58]) mediates essential interactions with the recombinase Rad51[50,52], its sequence shows a high degree of divergence, sharing less than 40% identity with the corresponding sequence in the closely related yeast *K. lactis*. Furthermore, the entire domain is predicted to be mainly disordered (Fig. 1B), as opposed to the N-terminal domain which is highly structured (Supplementary Fig. 1A). To assess the extent of disorder in this domain, we expressed the Rad52 (206–471) segment with uniform $^{15}$N (or $^{15}$N-$^{13}$C) labeling and probed its residual structure using Nuclear Magnetic Resonance (NMR) spectroscopy. As anticipated, the observed $^{15}$N-$^1$H spectrum was characteristic of a disordered protein, exhibiting reduced spectral dispersion of proton amide chemical shifts (Fig. 1C, gray spectrum). The majority of resonances were successfully assigned (Supplementary Fig. 1B) enabling the calculation of a chemical shift index for alpha carbons along the peptide chain. This analysis confirmed the disordered nature of the entire segment, with no evidence of residual secondary structure (Fig. 1D).

To identify the region of this domain that interacts with Rad51, full-length unlabeled Rad51 was added to the uniformly $^{15}$N labeled Rad52 (206–471) sample in a 1:1 ratio (Fig. 1C, blue spectrum). This resulted in a significant decrease in intensity of the NMR signal of approximately 85 residues located in the central region of the construct (Fig. 1E). To validate this observation, two shorter segments, covering the central region of Rad52 C-terminus Rad52 (295–408) and Rad52 (295–394), were isolated for further analysis. Consistent with the unfolded nature of these segments, nearly all resonances in the spectra of the short constructs (295–408 and 295–394) overlay with those of the longer construct (206–471) (Supplementary Fig. 1C). The addition of full length unlabeled Rad51 to $^{15}$N-labeled samples of the two shorter fragments confirmed that the Rad52 segment, ranging from residue 310 to 395, is involved in the interaction with Rad51 (Supplementary Fig. 1E). Interestingly, this region exhibits higher sequence conservation compared to the full C-terminal segment. When restricted to fungi, the alignment reveals three distinct motifs (Fig. 1F, higher panel): motif 1 (from residue 312 to 322, which contains the previously described FVTA motif[52]), motif 2 (from residue 337 to 367) and motif 3 (from residue 371 to 392, which contains the previously described YEKF motif[50]). This conservation is even higher in an alignment focused on *Saccharomyceae* (Fig. 1F, lower panel). Taken together, our data demonstrate that a large segment of 85 residues within the disordered C-terminal domain of Rad52, encompassing three motifs conserved in fungi, mediates the interaction with Rad51.

### Rad52-Cter dissociates Rad51 oligomers and binds the monomeric form

While Rad51 in solution is primarily oligomeric and elutes at the void volume of the size exclusion chromatography (SEC) column (1.04 mL), incubation of Rad51 with the Rad52 (295–394) or Rad52 (295–408) fragments at a 1:1 ratio enabled the isolation of a stable complex by

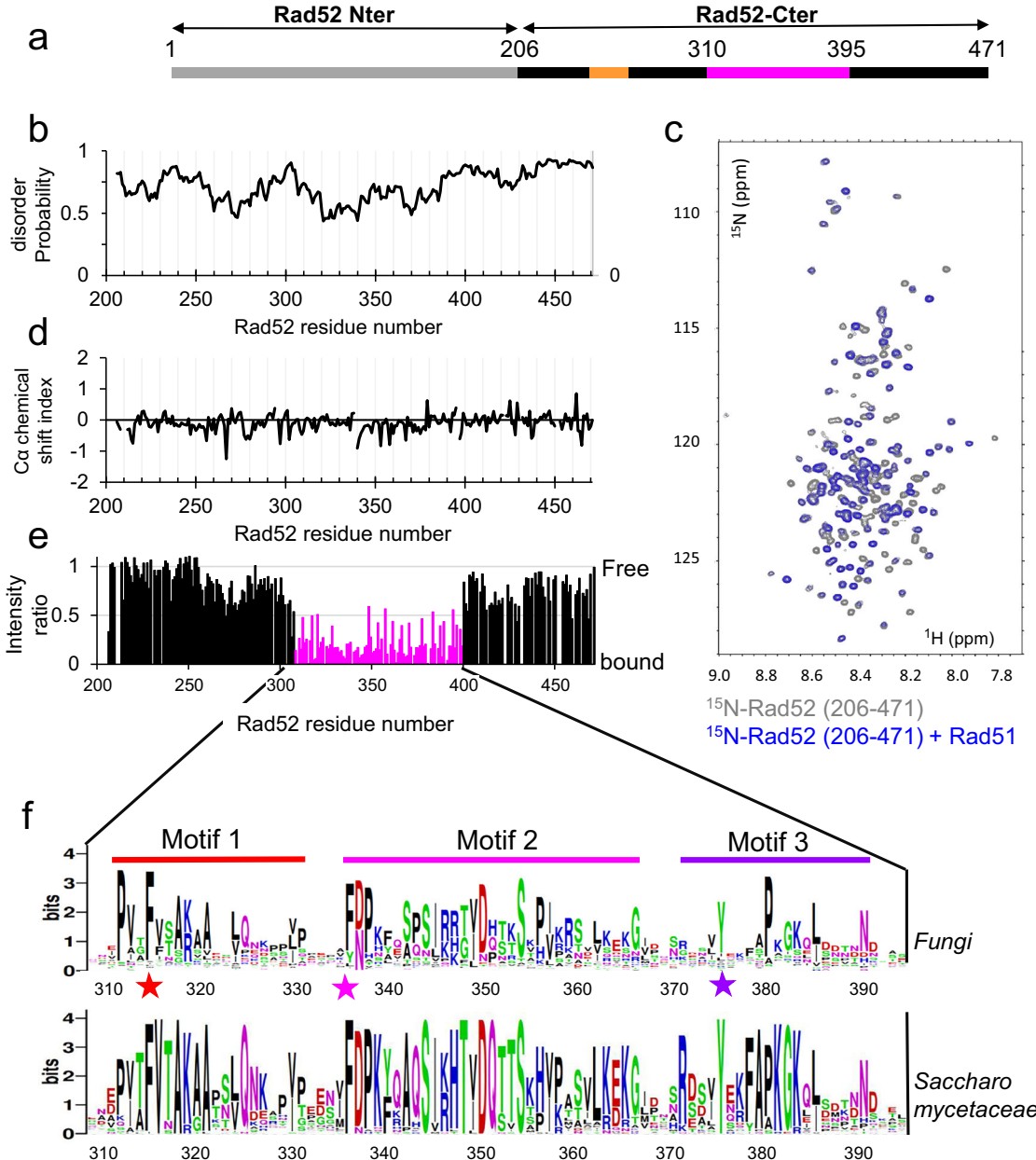

**Fig. 1 | Rad52 C-ter is disordered and interacts with Rad51 with a central region of 85 residues. a** General organization of Rad52 domains. The C-terminal domain is shown in black, with the region interacting with Rad51 in magenta, and the RPA binding region in orange[51]. **b** Predicted disorder of Rad52-Cter domain (206-471) (see Materials and Methods). **c** [1]H-[15]N SOFAST-HMQC spectrum of the uniformly [15]N labeled Rad52-Cter domain (206-471) alone in gray and in blue, after addition of equimolar amount of unlabeled Rad51. **d** Cα chemical shift index calculated for all assigned residues of the Rad52-Cter domain (206-471). The values close to zero of this index show that this domain is fully disordered, consistent with the predictions. **e** Mapping of the interaction between Rad52-Cter domain (206–471) and Rad51, using the intensities ratio (I/I0), where I and I0 are the intensity of the signals [1]H-[15]N SOFAST-HMQC spectra before and after addition of Rad51, respectively. **f** Sequence Logo of the central region of the Rad52-Cter domain (310–395) generated with homologues of Rad52 in fungi or in *Saccharomycetaceae*. F316 (Anchor 1), F337 (Anchor 2), and Y376 (Anchor 3) located in each conserved motif and mutated in this study are highlighted with a star. Disorder prediction, Cα chemical shift index and intensities ratio are provided in the Source Data file.

SEC. These results are fully consistent with previous analyses of this complex by SEC[52]. This complex corresponded to the major peak with elution volumes of 1.33 mL and 1.31 mL respectively (Fig. 2A, Supplementary Fig. 2A). Additionally, a minor peak (between 1.1 mL and 1.25 mL elution volume) containing Rad51 oligomers but no Rad52, was observed with lower intensity compared to the major peak (Fig. 2A, Supplementary Fig. 2A). SAXS (small angle X ray scattering) measurements combined with SEC and peak deconvolution, allowed the determination of an experimental molecular weight of 169 kDa

corresponding to four Rad51 monomers for the minor peak. In contrast, a molecular weight of 68 kDa, consistent with a 1:1 Rad51-Rad52C-ter complex (calculated molecular weight of 56 kDa), was measured for the major peak (Fig. 2B, Supplementary Fig. 2B). For this complex, the position of the maximum for the dimensionless Kratky plot was shifted to higher values on both the y- and x-axis compared to the position of the expected maximum for a fully globular protein (Supplementary Fig. 2C), indicating that the complex is extended, mainly folded, but retains significant flexibility.

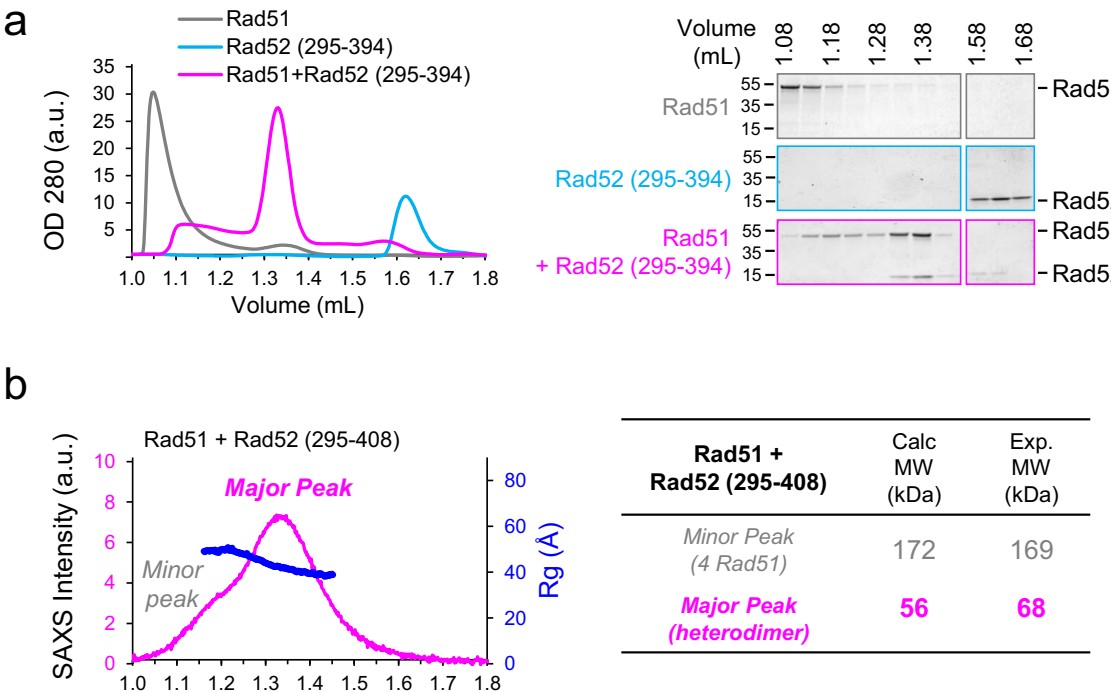

**Fig. 2 | Rad52 C-ter forms a 1:1 complex with Rad51 and prevents Rad51 oligomerization. a** Left panel: SEC profile analysis of Rad51, Rad52 (295–394) and Rad51+Rad52 (295–394) in a 1:1 ratio. Right panel: SDS-PAGE analysis of fraction from SEC profile in left panel revealed with coomassie blue. Position of molecular weight markers are indicated (in kDa), images representative of 2 independent experiments. Uncropped blots are provided in the Source Data file. **b** SAXS data for the 1:1 complex between Rad51+Rad52 (295–408).

Left panel: Series intensity (magenta, left axis) vs. elution volume, and, if available, radius of gyration (Rg) vs. elution volume (blue, right axis). The profile includes a major and a minor peak. The deconvolution of these peaks is shown in Supplementary Fig. 2B. Right panel: Molecular weight calculated with SAXS data (Exp. MW) compared to theoretical MW (Calc. MW) for the major and minor peaks. The full dataset is provided in the Source Data file.

## Rad52 (310–394) folds upon Rad51 binding burying a large conserved surface

Using the AlphaFold2 multimer software (AF2), we generated a model of the complex comprising one copy of full-length Rad51 and the Rad52 (295–408) segment (Fig. 3A, Supplementary Fig. 3A, B). The model was predicted with high confidence (see Materials and Methods). A significant portion of Rad52 (295–408) is predicted to fold upon binding to the C-terminal domain of Rad51 (77–400), covering a broad surface area and forming numerous interface contacts (total interface of 3003 Å², Supplementary Fig. 3C, Supplementary Table 1). Remarkably, the Rad52 residues implicated in the interface correspond exactly to the region (310–394) identified by NMR. To further evaluate the accuracy of this model, we tested its compatibility with the SAXS curve. By incorporating flexibility only into the N-terminal tail of Rad51, which is predicted to be flexible, and in the hinge between the N-terminal four-helix domain and the C-terminal domain (Supplementary Fig. 3D, black arrows), we generated a best-fit model that aligned closely with the experimental data ($\chi^2$ values ranging from 2.50 to 2.94) (Supplementary Fig. 3D, E). Superimposing the five generated models that all fit correctly the experimental SAXS curve, revealed no unambiguous orientation between the N-terminal domain of Rad51 and the C-terminal domain (77–400) (Supplementary Fig. 3F), indicating that the orientation of the N-terminal domain of Rad51 could remain flexible in the complex with Rad52. Taken together, our data demonstrate that a large segment of the Rad52 C-terminal tail (residues 310–394) binds to the C-terminal region of Rad51 (residues 77–400), forming a soluble 1:1 complex.

## Mutagenesis of residues predicted to be critical confirms the model

As already mentioned, the segment 310-394 of Rad52 shows significant sequence conservation in fungi (Figs. 1F, 3A, left panel). Reciprocally, the surface of Rad51 interacting with Rad52 is more conserved compared to the rest of the protein (Fig. 3B). Based on the predicted structure of the complex and residue conservation, we identified a set of key residues in Rad52 that are buried upon interaction and conserved, and thus predicted to be critical for the interaction with Rad51 (F316, V317, T318, A319, V325, F337, T349, V350, S355, V358, Y376, P381, Fig. 3A, right panel). These residues were mutated, and their ability to interact with Rad51 in yeast cells was assessed by co-immunoprecipitation (co-IP). For comparison, we also mutated residues predicted to be less critical for binding (K340, L363, S374) (Supplementary Fig. 3G, H). Single mutations of residues buried more than 40% led to a severe defect in Rad51-Rad52 co-IP. The plot (Fig. 3C) showing the percentage of interaction loss measured by co-IP as a function of the residue surface buried upon binding demonstrated a significant correlation (R = 0.76). We conclude from all the analyses presented above that the AlphaFold model, obtained with high confidence scores and fully compatible with our NMR, SAXS, and mutagenesis data, can be considered a reliable model.

We next evaluated the resistance of seven of these mutants to increasing doses of MMS (Fig. 3D, upper panels). Interestingly, mutants that exhibited a complete loss of interaction with Rad51, as assessed by co-IP, were all sensitive to MMS, in varying degrees, highlighting the functional importance of this interaction. Consistently, K340E and

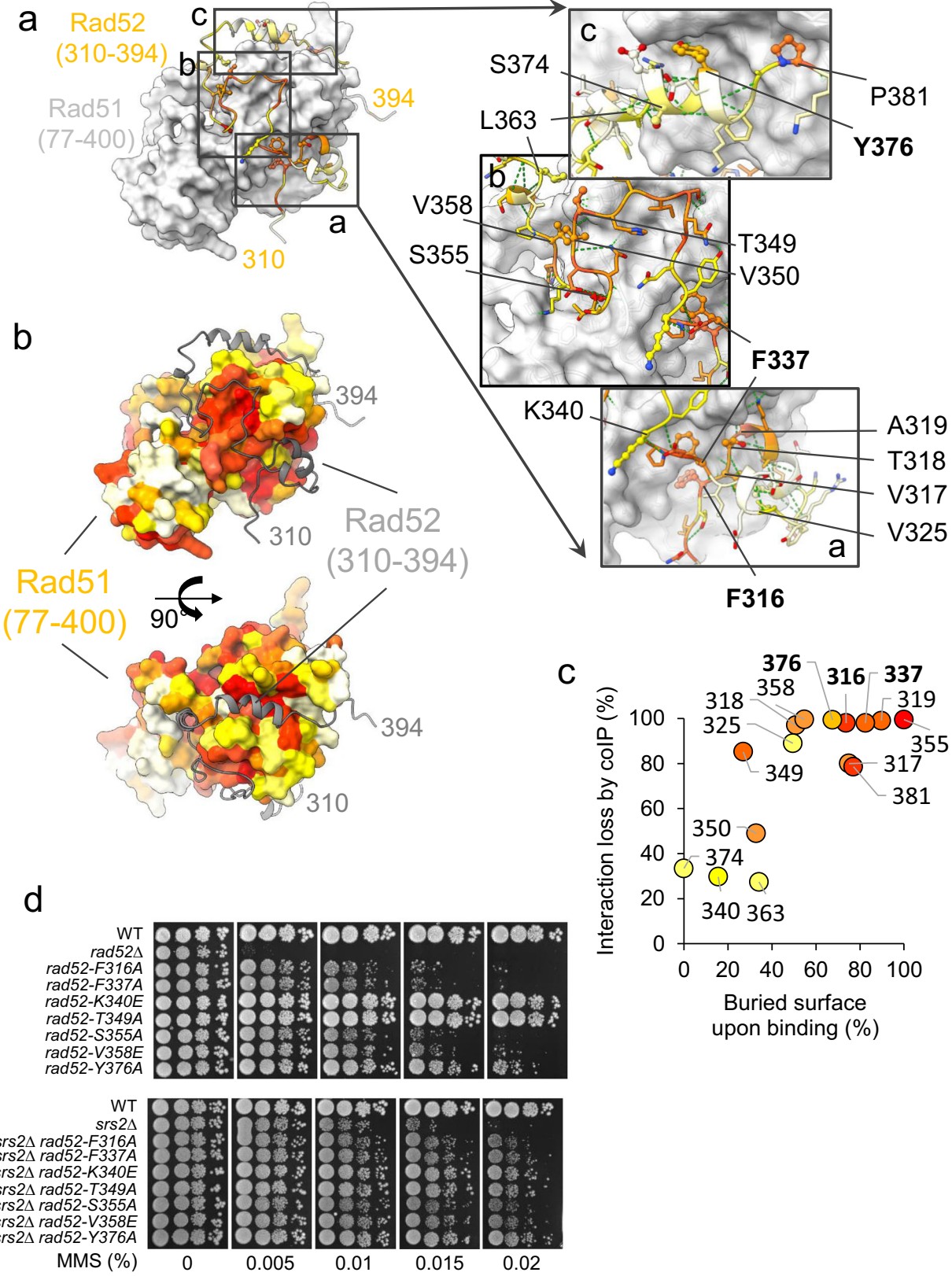

T349A mutations, which showed a partial loss of interaction with Rad51 (30% and 85%, respectively), displayed sensitivity levels comparable to wild-type (WT) cells. Note that the mutants fully defective in Rad51 interaction are less sensitive to MMS than the complete deletion of *RAD52* (*rad52Δ*), indicating that some activities of Rad52 are not affected in these mutants.

## Rad52 (310–394) competes with Rad51 oligomerization

The 310–394 region of Rad52 contains three motifs with significantly sequence conservation (Fig. 1F). We identified three key aromatic residues, F316, F337, and Y376 located in these motifs, and critical for Rad51 interaction in cells, as shown by coIP experiments (Fig. 3C, Supplementary Fig. 3G, H). These residues are entirely buried upon

**Fig. 3 | Rad52 (310–394) folds upon binding Rad51 burying a large conserved surface. a** Left panel: Best AlphaFold2 model of the complex between Rad51 (77–400) (gray surface) and Rad52 (310–394) (colored according to sequence conservation: from red for the highest conservation to white). Three boxes highlight zoomed-in regions shown in the right panels. Right panels: the side chains of Rad52 residues interacting with Rad51 are depicted as sticks, with mutated residues represented in ball-and-stick. Labels point to the C-alpha carbons of the mutated residues. Green dashed lines highlight hydrogen bonds. **b** AlphaFold2 model of the Rad51 (77–400) complex with its surface colored according to sequence conservation and Rad52 (310–394) shown as a gray cartoon. Two perpendicular orientations are provided in the upper and lower panels, respectively. **c** Interaction loss (%) plotted as a function of the surface area of each residue buried upon interaction. Interaction loss was determined by the intensity ratio between mutant and WT Rad52 co-immunoprecipitated with Rad51. **d** Serial 10-fold dilutions of haploid strains bearing different *RAD52* mutations of residue involved in the interaction, spotted onto rich medium (YPD) containing different MMS concentrations. Upper panel: *SRS2* background, lower panel: *srs2Δ* background. Source data are provided in the Source Data file.

interaction with Rad51 and correspond to anchor residues: F316 (Anchor 1) and F337 (Anchor 2) bind the Rad51 multimerization surface, competing with oligomer formation (Fig. 4A, B, Supplementary Fig. 4A, Supplementary Table 2,). F316 is part of the FVTA motif, where mutations to any residue disrupt binding (Fig. 3C, Supplementary Fig. 3G, H). Interestingly, this motif adopts the same binding mode as the FVTA motif which mediates Rad51 oligomerization (F144-VTA). In Rad52, the motif extends with F337 folding back near F316 (Fig. 4A, B, Supplementary Fig. 4A). In contrast, Y376 (Anchor 3) is part of the third conserved motif and located in a helix that folds upon Rad51 interaction but does not compete with its oligomerization surface (Fig. 4A, B).

To test the relative contribution of individual motifs to the Rad51-Rad52 (295–394) interaction, we used NMR to analyze uniformly labeled [15]N Rad52 (295–394) mutated for each of the three anchor residues (F316A, F337A, or Y376A) as well as a triple mutant (noted FFY_AAA). The loss of signal intensity after the addition of Rad51 was measured in the same conditions as for the WT Rad52 fragment. We observed a significant reduction in binding for all three mutants, not only in the mutated motifs but across the entire fragment (Fig. 4C, Supplementary Fig. 4B), highlighting the cooperative nature of all anchor interactions. This property was further confirmed by gel filtration, as these mutants were no longer able to dissociate Rad51 oligomers (Fig. 4D, Supplementary Fig. 4C).

In summary, we demonstrated that the C-terminal segment (310-394) of Rad52 binds cooperatively Rad51. The F316VTA motif of Rad52 extends to include F337 (anchors 1 and 2) and competes with the Rad51 dimerization interface, providing the molecular basis for the biochemical property of this region of Rad52 to dissociate Rad51 oligomers. These motifs are conserved in fungi, suggesting that this binding mode is preserved across the kingdom. Additionally, the Rad51-Rad52-Cter interaction surface involves residues outside the Rad51 oligomerization surface, such as Y376 allowing possible residual binding with Rad51 filament, which could occur with the full protein.

## Rad52 anchors 1 and 2 are predominantly involved in Rad51 filament formation in vivo

The mutation of each of the three anchor residues leads to a severe loss of interaction with Rad51 both in vitro, as observed by NMR and SEC, and in vivo as shown with the full-length Rad52 protein by co-IP (Fig. 3C, Supplementary Fig. 3G, H, Fig. 4C, D). However, *rad52-F316A* and *F337A* (anchors 1 and 2) exhibit higher sensitivity to MMS (Fig. 3D) compared to *rad52-Y376A* (anchor 3), suggesting a more pronounced impact on Rad51 filament assembly. Previously, we showed that resistance to MMS is restored in *rad52-Y376A* cells in the absence of the Srs2 translocase, which efficiently displaces Rad51 from ssDNA[41]. Based on this observation, we concluded that Rad52 binding to Rad51 through this residue (located in the conserved Motif 3) confers protection to the Rad51 filament against destabilization by Srs2. In this study, we observe the same genetic relationship between Rad52 mutations in Motifs 1 and 2 (*rad52-F316A, F337A, S355E* and *V358E*) and *srs2Δ* (Fig. 3D, lower panel, Fig. 1F for the delimitation of the conserved Motifs). This indicates that the Rad51 filament assembly is not totally impaired in these mutants and that the binding of the entire Rad52 (310–394) segment is essential to prevent the destabilization of Rad51 filaments

by Srs2. Alternatively, the absence of Srs2 could compensate the poor efficiency of these mutants to assemble Rad51 filaments.

Exposure to γ-rays also shows a stronger loss of viability of *rad52-F316A* (Anchor 1) and *F337A* (Anchor 2) mutants compared to *Y376A* (Anchor 3) (Fig. 5A, left panel). Similarly to MMS exposure, the combination of *srs2Δ* with these three anchor mutants rescued γ-ray sensitivity (Fig. 5A, right panel). *Rad52-F316A* and to a lesser extent *F337A* mutants are also more sensitive than *Y376A* to the formation of a single HO-induced DSB that can be repaired by gene conversion, using a donor sequence located on another chromosome[59] (Fig. 5B). In addition, we used a strain carrying an HO-inducible DSB site that can only be repaired by the Rad51-independent mechanism of single strand annealing (SSA) to assess the ability of Rad52 mutants to assemble stable Rad51 filaments. In this strain, Rad51 is not required for DSB repair, but Srs2 is essential to remove Rad51 filaments from the ssDNA generated at the DSB, which would otherwise block repair completion[60,61]. In this context, the mutation of Anchor 3 behaves like a WT (Supplementary Fig. 5A). In contrast, *rad52-F316A* and *F337A* mutant cells exhibit significant sensitivity, but restore the viability of *srs2Δ* cells, as we previously found for *rad52-Y376A*[41] (Supplementary Fig. 5A). These results confirm the destabilization of Rad51 filaments in Anchor 1 and 2 mutant cells.

To assess the defects of the three anchor mutants at the molecular level, we measured the recruitment of Rad51, RPA, and Rad52-FLAG proteins by ChIP in cells expressing *radS2-FLAG, radS2-F316A-FLAG,* or *radS2-F337A-FLAG* alleles, as we previously did for *rad52-Y376A-FLAG* cells in the same strain[41] (Fig. 5C, Supplementary Fig. 5B). Quantitative PCR using primer sets targeting DNA sequences 0.6 kb upstream of the DSB site, 4 h after DSB induction, showed an increased relative enrichment of RPA, Rad52-FLAG and Rad51 at the DSB site compared to the 0 time point and the uncut *ARG5,6* locus in WT cells[41]. No significant differences were observed in the recruitment of RPA, Rad52-F316A-FLAG or Rad52-F337A-FLAG compared to Rad52-FLAG cells (Supplementary Fig. 5B). The binding of RPA and Rad52-FLAG mutant proteins was also unaffected in Srs2-deficient cells. Conversely, Rad51 loading was reduced by 33- and 21-fold in *rad52-F316A* (Anchor 1) and *rad52-F337A* (Anchor 2) mutant cells, respectively, compared to *RAD52-FLAG* cells. This important reduction observed for Anchor 1 and 2 mutants is significantly greater than the 2-fold reduction observed for Anchor 3 mutants (Fig. 5C). Furthermore, deletion of the *SRS2* gene in Anchor 1 and 2 mutants resulted in a 13-fold and 8-fold reduction in Rad51 recruitment, respectively, compared to *srs2Δ* cells. These results suggest that Anchor 1 and 2 mutants are severely defective in Rad51 filament formation. In contrast, *SRS2* deletion in the Anchor 3 mutant fully restored Rad51 recruitment, indicating that this mutant efficiently mediates Rad51 filament formation but is poorly able to compete with Srs2.

To further evaluate the impact of the Rad51-Rad52 interaction on Rad51 filament formation in living cells, we introduced mutations in each of the three Rad52 Anchors residues into a strain expressing a functional GFP-tagged version of Rad51 as the only copy of Rad51[19]. We monitored the number, intensity and shapes of Rad51 structures formed after inducing a DSB at the *lys2* locus by expressing the *I-SceI* endonuclease (Fig. 6). As previously reported, more than 80% of WT cells form either filaments (>50%) or bright foci 4 h after Isce-I induction([19] and Fig. 6A). In contrast, in strains expressing *rad52*

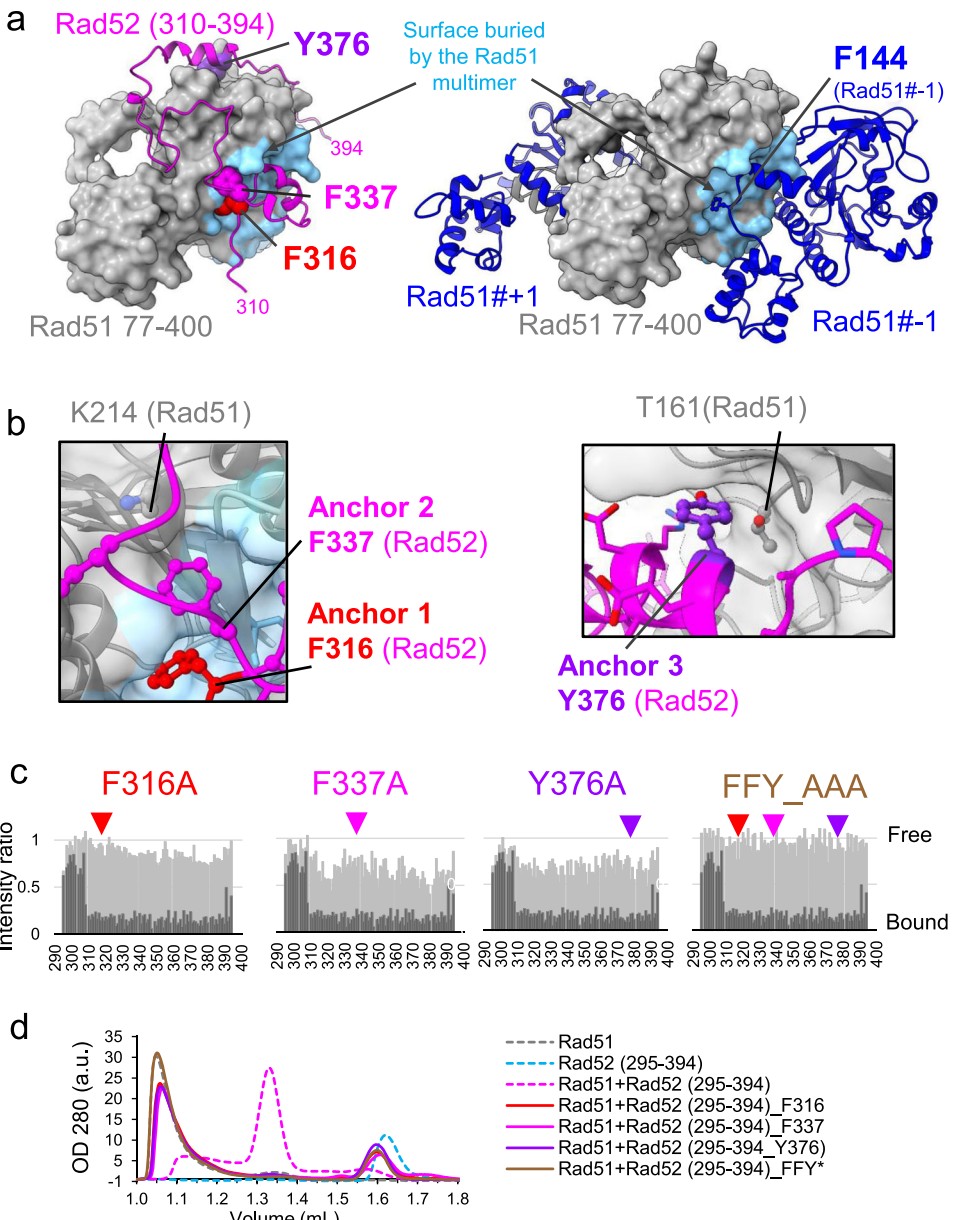

**Fig. 4 | Rad52 (310–394) competes with Rad51 oligomers through the cooperative binding of several anchors. a** Left panel, best AlphaFold2 model of the complex between Rad51 (77–400) (gray surface) and Rad52 (310–394) (colored in magenta). The three key residues mutated are highlighted with balls. Right panel, structure of three Rad51 monomers assembled in a multimer (PDB code 1SZP). The central protein is truncated (Rad51 (70–400) and shown as a gray surface), the two other Rad51 molecules (#-1 et #+1) are shown as cartoon in blue. Residue F144 side chain is highlighted. For both panels, the Rad51 surface buried upon formation of the Rad51 filament is colored in blue. **b** Zoomed-in of residues F316 and F337 (left panel) and Y376 (right panel). The Rad51 surface is shown with transparency to show the position of K214 and T161 from Rad51 contacting F337 and Y376, respectively. The Rad51 surface buried upon formation of the Rad51 filament is colored in blue. **c** In light gray, mapping of the interaction between Rad52-Cter domain mutants (295–394) and Rad51, using the intensities ratio (I/I0), where I and I0 are the intensity of the signals $^1$H-$^{15}$N SOFAST-HMQC spectra before and after addition of Rad51, respectively. For comparison, values for the WT are indicated in dark gray. **d** SEC profile analysis of Rad51, Rad52 (295–394) and Rad51 + Rad52 (295–394) WT and mutants in a 1:1 ratio. Source data are provided in the Source Data file.

mutants, most cells show weak Rad51 foci with intensity 5 to 10-fold weaker than the one observed in wild-type cells and no detectable structures in 16% to 34% of cells (Fig. 6A, B, Supplementary Fig. 6). The relative severity of the phenotypes of the three mutants is in good agreement with Rad51 ChIP (Fig. 5C), and all functional tests presented above including MMS and γ-rays sensitivity as well as gene conversion efficiency, with the *rad52-Y376A* (Anchor 3) mutant showing the weakest phenotype and the *rad52-F316A* (Anchor 1) mutant showing the strongest phenotype. Deletion of *SRS2* in these strains rescued the

ability of Rad51 to form detectable structures, although the proportion of cells forming filaments was smaller and filaments were weaker than those observed in WT or *srs2Δ* strains. Consistently with our ChIP data, deleting *SRS2* in *rad52-Y376A* strain rescues foci intensity to levels comparable to WT and *srs2Δ* strains. In contrast, foci intensities in the *rad52-F316 srs2Δ* and *rad52-F337 srs2Δ* remain significantly weaker than in the *srs2Δ* strain although higher than in the *rad52-F337* and *rad52-F337* single mutants. Again, the relative capacity of the double mutants to form Rad51 structures agrees with ChIP data and functional tests

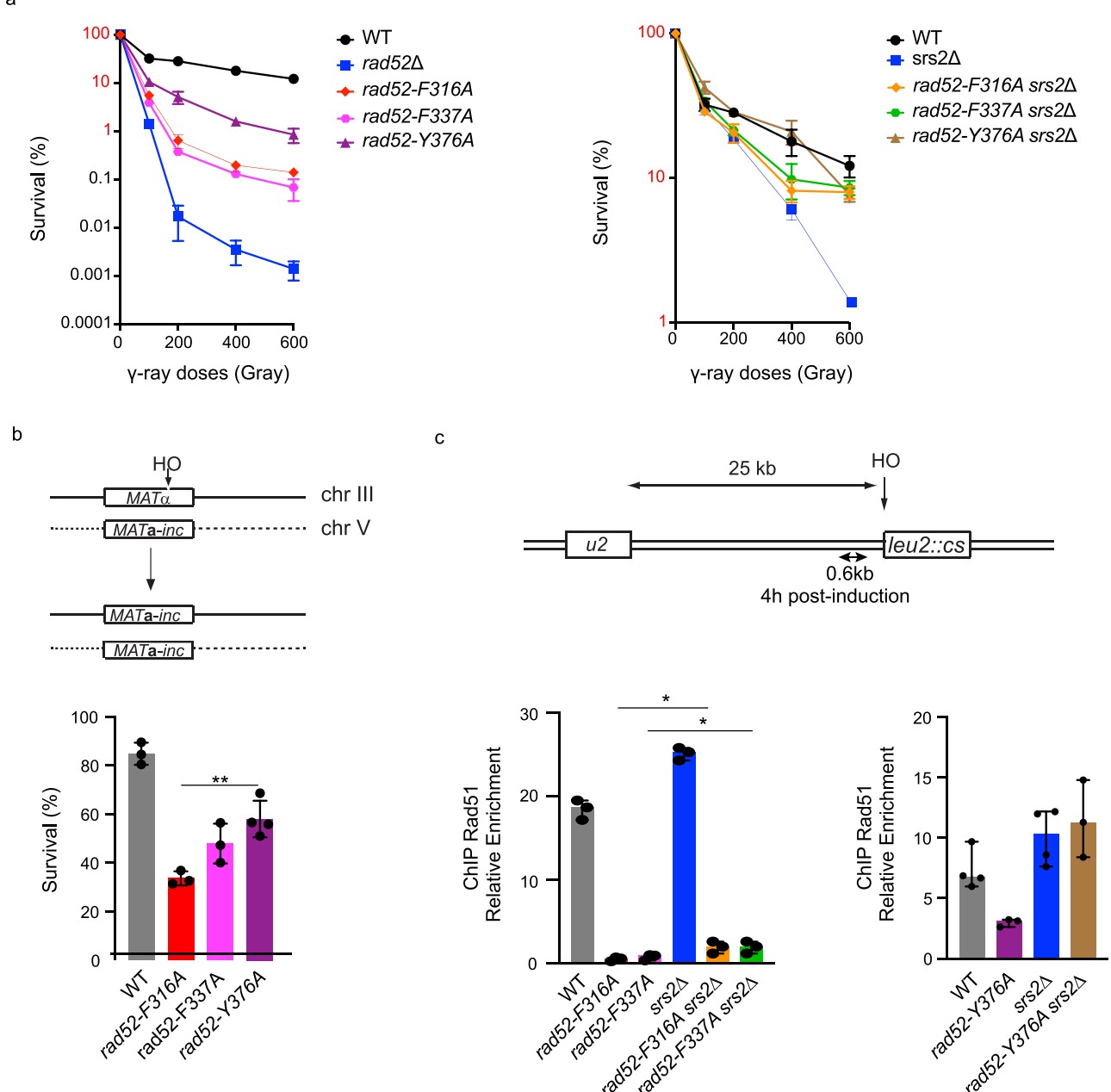

**Fig. 5 | Anchor 1 and 2 are more important for Rad51 filament formation than Anchor 3. a** Survival curves of haploid cells with the indicated genotypes exposed to γ-rays. Survival from 1 to 100% is labeled in red to underline the difference of scale displayed on the left and the right panels. Each data points indicates the mean ± SEM. WT $n = 4$, $rad52\Delta$ $n = 6$, $rad52$-$F$-$316A$, $rad52$-$F337A$, $srs2\Delta$, $rad52$-$F316A$ $srs2\Delta$, $rad52$-$F316A$ $srs2\Delta$ $n = 3$. Results from $rad52$-$Y376A$ and $rad52$-$Y376A$ $srs2\Delta$ are from ref. 41 (CC BY 4.0: https://creativecommons.org/licenses/by/4.0/). **b** Survival rates of cells suffering an HO-induced gene conversion between *MAT* ectopic alleles. Bars represent the mean ± SD ($n = 3$). Results from $rad52$-$Y376A$ are from[41]. (**$p = 0.0033$, two-tailed unpaired $t$ test). **c** ChIP analysis shows that $rad52$-$F316A$

(Anchor 1) and $rad52$-$F337A$ (Anchor 2) highly impact Rad51 recruitment at a HO-induced DSB and the low dependency on Srs2. Upper panel: schematic of the HO-induced SSA repair system used. Lower panel: Rad51 ChIP relative enrichment was assessed at 0.6 kb from the DSB site 4 h after HO induction. Data are presented as the median and error bars represent the minimum and maximum values ($n = 3$). *Statistical analysis shows that deleting *SRS2* in the $rad52$ mutants increase slightly but significantly the recruitment of Rad51 ($rad52$-$F316A$ versus $rad52$-$F316A$ $srs2\Delta$, $p = 0.029$, $rad52$-$F337A$ versus $rad52$-$F337A$ $srs2\Delta$, $p = 0.06$; two-tailed unpaired $t$ test). Already published data obtained with $rad52$-$Y376A$ (Anchor 3) are shown for reference [41]. Source data are provided in the Source Data file.

presented above (Fig. 5). These data confirm the importance of Rad51-Rad52-Cter interaction for the formation of long and stable Rad51 filaments with a predominant role of Anchors 1 and 2, while mutation in Anchor 3 shows a less severe phenotype both in the presence or absence of Srs2.

Taken together, our results show that all three anchors contribute to Rad51 filament formation, with anchors 1 and 2 playing the major role by interacting with the polymerization interface of Rad51.

**Mutants targeting Rad51 residues in contact with Rad52 confirm the functional role of the Rad51-Rac52Cter**

To provide additional support to our conclusions, we mutated Rad51 residues at the Rad52-Rad51 interface facing Rad52 Anchor residues. Anchor 1 faces the surface of polymerization of Rad51, mimicking the Rad51-F144 residue (Fig. 4A, right panel, Supplementary Fig. 4A). Consistent with this, the $rad51$-$F144D$ mutation disrupts the Rad51-Rad51 interaction, as shown by yeast two hybrid (Y2H) analysis

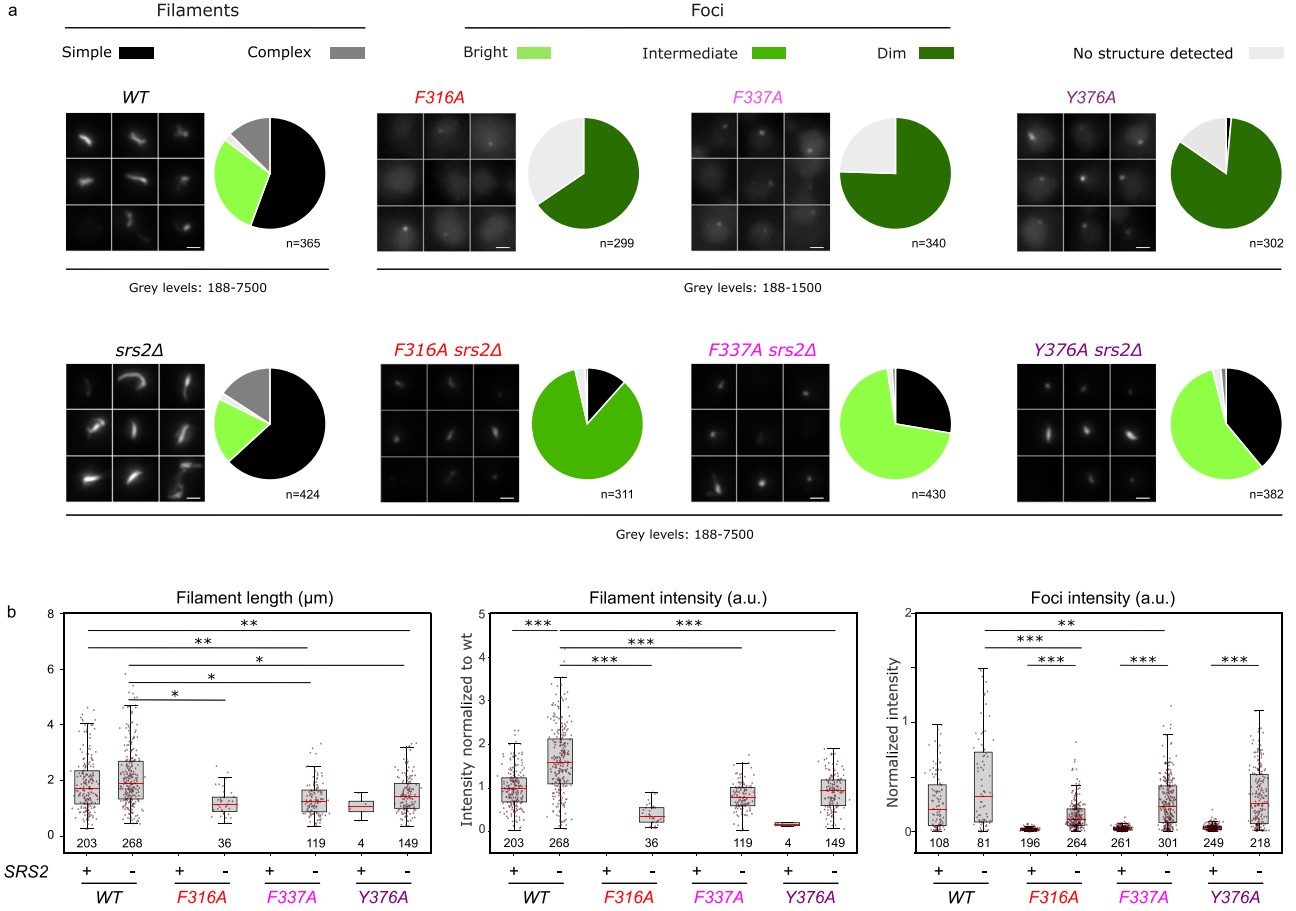

**Fig. 6 | Impact of Rad52-Rad51 binding anchor mutants on Rad51 structure formation in living cells. a** Representative images of Rad51-iGFP 4 h after Isce-I induction in WT, *srs2Δ*, *rad52* single mutants or *rad52 srs2Δ* double mutants. Maximum Z-projection is applied on fluorescent images. Minimum and maximum fluorescence intensities are indicated. Scale bar 1 μm. Pie charts show the proportion of cells with different Rad51 structures as indicated. Bright foci correspond to mean foci intensities > 80% of WT, intermediate to mean values between 20 and 80% of WT, and dim to mean values < 20% of WT. Data are from at least 3 independent experimental replicates. The number of cells analyzed (n) is indicated below each pie chart. **b** Comparison of Rad51 foci intensities, filament intensities and lengths in WT and mutant strains as indicated. The number of structures analyzed is indicated below each box. Intensities are normalized to the mean

intensity of WT Rad51 filaments. On each box, the central mark indicates the median, and the bottom and top edges of the box indicate the 25th and 75th percentiles, respectively. The whiskers extent corresponds to the adjacent value, which is the most extreme data value not considered an outlier (above 75th percentile +1.5 times interquartile range or below 25th percentile −1.5 times interquartile range). Statistical test: logistic regression with binomial distribution, t-statistic on coefficients corrected for multiple comparison with the False Discovery Rate. (See materials and methods; ***$P < 0.001$, **$P < 0.01$, *$P < 0.05$). Exact *p*-values for all comparisons are presented in Supplementary Fig. 6. Source data are provided in the Source Data file. Microscopy data as well as the complete Source Data file are available in Zenodo: https://zenodo.org/records/15149086.

(Supplementary Fig. 7A). Additionally, the sensitivity of *rad51-F144D* mutant cells to MMS is identical to that of the *rad51Δ* mutant (Supplementary Fig. 7B), showing that this residue is essential for Rad51 filament formation. Therefore, it will not be possible to mutate residues of the Rad51-Rad51 interface to analyze the interaction with Rad52. We then mutated Rad51-K214 which, based on our structural model of the Rad51-Rad52 complex, interacts with Rad52-F337 (Anchor 2) but does not belong to the Rad51 polymerization interface (Fig. 4B, left panel). As predicted by our structural model, *rad51-K214M* abolishes most of the interaction with Rad52 as seen by co-IP, while the stability of the protein is not affected (Supplementary Fig. 7C). In contrast, yeast two-hybrid analysis shows that the interaction with Rad51 and the Rad55-Rad57 complex is not affected (Supplementary Fig. 7D). As for *rad52-F337A* (Fig. 3D), *rad51-K214M* cells are very sensitive to MMS but this phenotype is not rescued by the deletion of *SRS2* (Supplementary Fig. 7E). Overall, these results confirm that interaction with Rad52 Anchor 2 is required for Rad51 filament formation.

In our structural model of the Rad51-Rad52 complex, Rad51-T161 contacts Rad52-Y376 (Anchor 3) (Fig. 4B, right panel). Consistently, the

mutant Rad51-T161R loses completely the interaction with Rad52, as observed by co-IP (Fig. 7A). In comparison with *rad51-K214M* cells, *rad51-T161R* cells shows a weaker sensitivity to MMS, comparable to that of *rad52-Y376A* cells (Figs. 7B and 3D)[41]. Moreover, this mutant leads to a complete suppression of the MMS sensitivity of Srs2-deficient cells, as observed for *rad52-Y376A*. However, *rad51-T161R* only partially suppresses the sensitivity of Srs2-deficient cells to γ-rays exposure (Fig. 7C), in contrast to what we observed for *rad52-Y376A* (Fig. 5A, right panel), suggesting that this mutation affect Rad51 filament stability to a lesser extent. Rad51-T161R binding to an HO-induced DSB is weakly affected compared with WT as observed by Rad51 ChIP, to the same extent as Rad51 in *rad52-Y376A* cells (2.8-fold compared to 2-fold, Fig. 5C). This reduction is entirely dependent on Srs2 activity (Fig. 7D). Consistently, the *rad51-T161R* strain displays a similar defect to form Rad51-iGFP filaments upon DSB as the *rad52-Y376A* (Fig. 7E, Supplementary Fig. 7F, G). Indeed, *rad51-T161R* strain shows mainly weak Rad51 foci or no detectable Rad51 structures following DSB induction. Similar to the *rad52-Y376A*, deleting *SRS2* in the

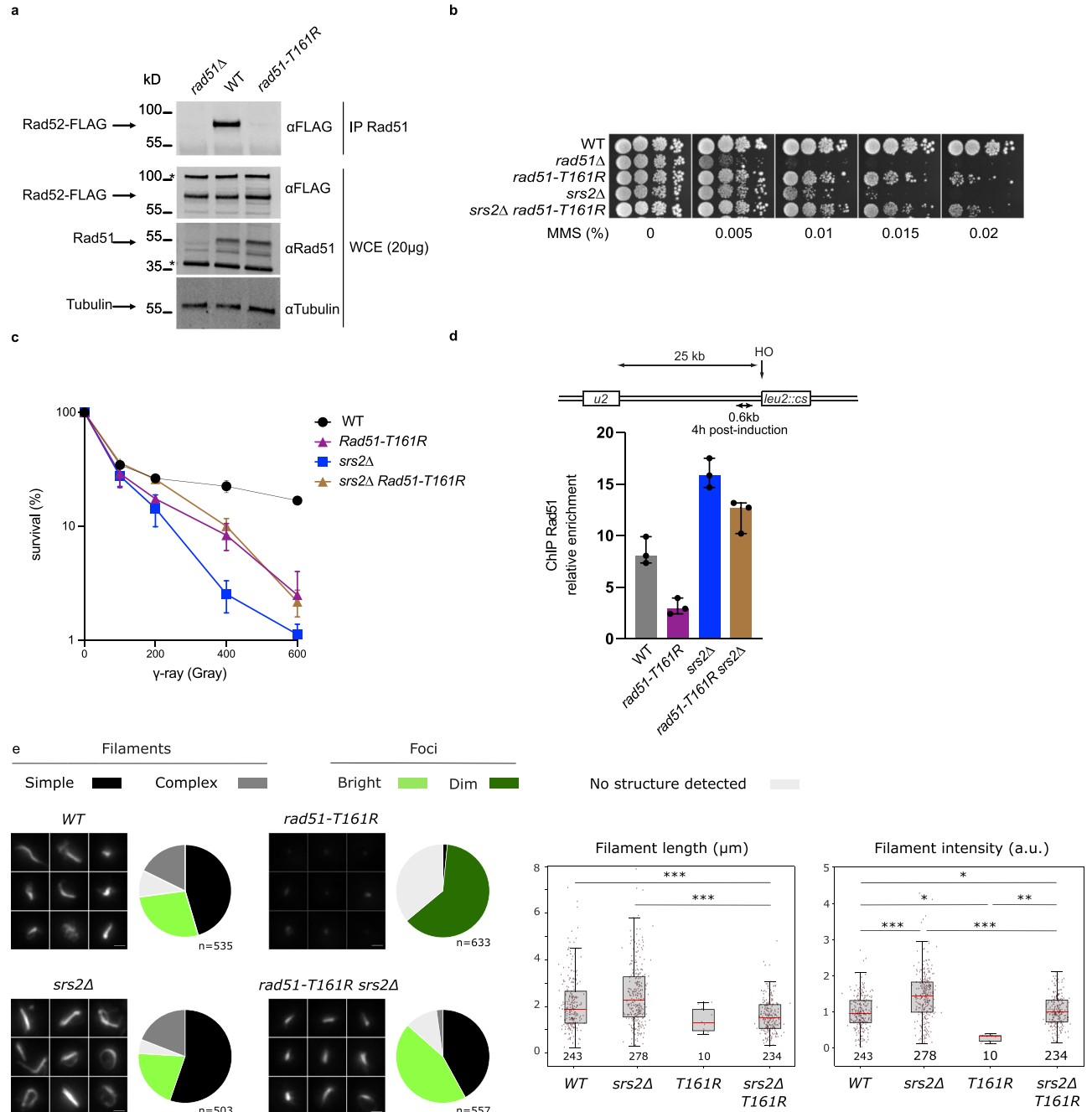

**Fig. 7 | rad51-T161R mimics rad52-Y376A. a** Co-IP experiments showing the loss of interaction between Rad51-T161R and Rad52. Rad51 was immunoprecipitated with a poly-clonal anti-Rad51 antibody (αRad51). The presence of Rad51 in the immunoprecipitated fraction (IP) cannot be detected because it migrates at the same level as the anti-Rad51 IgG used for the immunoprecipitation. However, the absence of Rad52-FLAG in the *rad51Δ* immunoprecipitate confirmed that the Rad52-FLAG signal observed is related to the Rad52–Rad51 interaction. Position of molecular weight markers are indicated (in kDa).* Unspecific bands. Uncropped blots are provided in the Source Data file. **b** Serial 10-fold dilutions of haploid strains with the indicated genotypes were spotted onto rich medium (YPD) containing different MMS concentrations. **c** Survival curves of haploid cells with the indicated genotypes exposed to γ-Rays. Each data points indicates the mean ± SD (*n* = 3). **d** ChIP was used to assess Rad51 relative enrichment at 0.6 kb from an HO cut site 4 h after HO induction. Data are presented as the median and error bars represent minimum and maximum values (*n* = 3). **e** Left panel: Representative images of Rad51-iGFP 4hrs after Isce-I induction in WT, *srs2Δ*, *rad51-T161R* or *srs2Δ rad51-T161R* double mutants. Maximum Z-projection is applied on fluorescent images. Scale bar 1 μm. Pie charts show the proportion of cells with different Rad51 structures as indicated (as in Fig. 6A). Data are from 3 independent experimental replicates. The number of cells analyzed (n) is indicated below each pie chart. Right panel: Comparison of Rad51 filament intensities and lengths in WT and mutant strains as indicated. The number of structures analyzed is indicated below each box. Intensities are normalized to the mean intensity of WT Rad51 filaments. Plots and statistical analyses are as in Fig. 6. Exact p-values for all comparisons are presented in Supplementary Fig. 7G. Source data are provided in the Source Data file. Microscopy data as well as the complete Source Data file are available in Zenodo: https://zenodo.org/records/15149086.

*rad51-T161R* strain rescues Rad51 foci intensity to levels comparable to WT and *srs2Δ* strains. Deleting *SRS2* partially rescues the ability to form filaments in 44% of the cells vs 63% in wild-type cells, although these filaments are shorter than the one formed in wild-type or *srs2Δ* strains.

These data further confirm the functional role of Rad52 anchor 3 association to Rad51 for the formation and stability of Rad51 filaments.

## Discussion

### A large Rad52-Rad51 interaction domain promotes Rad51 filament formation and stability

Our NMR studies revealed that the *S. cerevisiae* Rad52 C-terminal domain interacts with Rad51 through a large domain of 85 residues. Using AF2 modeling, we generated a model of the interaction in complete agreement with the NMR and SAXS data: the Rad52 residues involved in the interface correspond exactly to the region (310-294) identified by NMR. Co-IP experiments in vivo confirmed that most of the predicted residues are indeed involved in the interaction. Sequence alignments in fungi reveal three conserved motifs in the interaction region. The AF2 model suggests that motifs 1 and 2 bind the Rad51 multimerization surface and compete with oligomer formation. In contrast, the third conserved motif is located in a helix that folds upon Rad51 interaction, outside the multimerization surface. We show that Anchor 3 in this motif is important for Rad51 filaments to better resist Srs2, but is not essential for Rad51 filament formation, consistent with our previous observations[41]. In addition, mutations in the two other conserved motifs (Anchor 1 and Anchor 2) have a more pronounced impact on Rad51 filament formation. Therefore, we propose that these residues play a more critical role in this process. Mutations of all residues involved in the interaction with Rad51 result in significant sensitivity to DNA damage, at least partially dependent on Srs2, suggesting that each of them contributes to Rad51 filament formation to varying degrees and plays a role in filament homeostasis.

### Rad52 binding to Rad51 dissociates Rad51 oligomers to catalyze Rad51 filament formation acting as an assembly chaperone

SEC and SAXS analysis revealed that the C-terminal domain of Rad52 binds a Rad51 monomer. These findings indicate that Rad52 can dissociate Rad51 oligomers that spontaneously form in solution, as previously observed[52]. Our structural model provides a rationale for this property of Rad52 C-terminus: a segment spanning from Anchor 1 (F316) to Anchor 2 (F337) competes with the Rad51 multimerization surface. Mutating these anchors results in severe in vivo phenotypes, including high sensitivity to DNA damage agents, and poor Rad51 recruitment at DSBs, as demonstrated by ChIP and microscopy analysis.

At first glance, it might seem paradoxical that the residues directly competing with Rad51 oligomer formation play the most critical role in promoting Rad51 filament assembly. However, this is well documented characteristic of assembly chaperones, which are defined as factors that bind monomers to produce functional multimers but are not incorporated into the final complex[62]. These chaperones generally interact with sticky monomers surfaces, preventing aberrant associations. One prominent example is histone chaperones, a diverse group of proteins that interact with various histone surfaces, protecting them, and facilitating nucleosome formation in specific pathways[63,64]. Assembly chaperones have also been identified for other complexes, such as the proteasome and the ribosome[65,66], highlighting that molecular assembly chaperones are general feature in the formation of large macromolecular assemblies.

Notably, full-length Rad52 exists in solution as a homodecamer, and the estimated diffusion coefficients of Rad52 in living cells support the existence of multimeric forms of Rad52 in vivo[25,49]. This oligomerization ability is conferred by its N-terminal annealase domain, whose structure was recently solved[25] (Supplementary Fig. 1A). In that study, the authors proposed that the ssDNA-binding property of this domain, together with the association of Rad51 with the Rad52 C-terminal domain, could facilitate the transfer of Rad51 onto single-stranded DNA[25]. Rad52's ability to bind and solubilize Rad51 monomers through multiple C-terminal tails may facilitate the nucleation of two to five Rad51 monomers required on DNA to initiate further cooperative polymerization, as reported for its functional homolog, BRCA2[23,67-69]. In this manner, Rad52 could deliver Rad51 to DNA via its C-terminal motifs 1 and 2, especially by binding to the Rad51 multimerization surface, competing with premature oligomer formation and recruiting sufficient Rad51 monomers to promote Rad51 filament elongation (Fig. 8). The cooperative nature of Rad52 residues involved in Rad51 binding suggests that most of these interactions are weak, which may facilitate the dissociation of Rad52 from the Rad51 multimerization surface after Rad51 loading and promote Rad51 filament extension.

### Rad52 also binds Rad51 to efficiently compete with Srs2

In contrast, Rad52-Y376, part of the third conserved motif of the Rad51 binding site, is located in a helix that folds upon Rad51 interaction but does not compete with its oligomerization. As observed previously, mutation of this residue to alanine confers milder phenotype than *radad52-F316A* and *F337A* and the smaller decrease of Rad51 recruitment at DSBs as observed by ChIP is largely dependent on Srs2[41]. This was further confirmed here by the study of functional Rad51-iGFP filament formation at a IsceI-induced DSB by fluorescent microscopy. We have also shown in vitro that Rad52 remains associated with the Rad51 filament it helped to form on DNA previously coated with RPA[8]. Our structural model suggests that the third conserved motif containing Y376 does not compete with the Rad51 oligomerization interface on DNA and is therefore compatible with binding to Rad51 filaments. This configuration could explain the protection of the Rad51 filament against Srs2 activity (Fig. 8). However, we have observed a very important effect of the Rad52-Y376 mutation on the interaction with Rad51 by NMR with Rad52 C-terminus peptides and by co-IP with the full-length Rad52, as observed with Rad52-F316A and F337A. Why this mutation has a weaker effect on Rad51 filament formation remains to be determined. This might be linked to the ability of Rad52 to synergize with the Rad51 paralog complex Rad55-Rad57 and SHU to promote Rad51 filament formation[33]. Indeed, it is possible that the *rad52-Y376A* mutant has a better ability to cooperate with the Rad51 paralogs compared with the two other mutants. However, the very similar phenotypes of the *rad51-T161R* and *rad52-Y376* mutations, together with the fact that Rad51-T161 and Rad52-Y376 face each other at the interface, strongly support the idea that anchor 3 is particularly important to compete with Srs2 activity, while it is less critical for Rad51 filament formation. We speculate that the interaction through motif 3 would lead Rad52 to the 3'end of Rad51 filaments, competing with Srs2 and favoring the recruitment of another Rad51 monomer through another Rad52 C-terminus.

### Similarities and differences of Rad51 binding to Rad52 *vs.* BRCA2

A motif similar to the FVTA has also been identified in the BRC repeats of BRCA2 (F1524-HTA), the human functional homolog of Rad52, and binds RAD51 in a similar manner (Supplementary Figs. 4A, 8). BRCA2 has also been proposed to dissociate RAD51 oligomers[14,70-73] and to bring RAD51 monomer onto DNA to promote filament formation[26,27,30,74]. Direct competition with the FxxA motif involved in Rad51-Rad51 oligomerization is thus a conserved feature between both Rad52 and BRCA2.

Sequence analysis of the eight motifs of human BRCA2 reveals that the conserved region extends beyond the FxxA core, spanning approximately 25–30 residues (Supplementary Fig. 8A). The C-terminal extension of this region forms a helix that docks onto the RAD51 surface[14] (Supplementary Fig. 8B). AlphaFold2 predicts that this

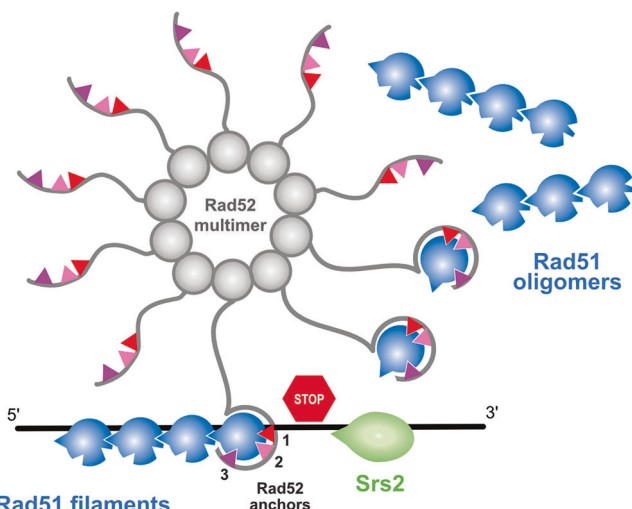

**Fig. 8 | Model depicting Rad52 assembly chaperone activity and protection against Srs2, mediated by the interaction of anchors 1, 2 and 3 with Rad51.** The N-terminal domain of Rad52 is represented as a gray sphere and assembles into a decamer, gathering ten unfolded C-terminal regions of Rad52, shown as wavy gray lines. The three Rad52 anchor residues, labeled 1, 2, and 3, are represented as colored triangles using the same color code as in Fig. 4–7. Each C-terminal tail can potentially bind a single Rad51 molecule (shown in blue), thereby dissociating Rad51 oligomers. The Rad51 motif involved in oligomerization (F144VTA) is represented as a blue triangular extension. Anchor 1 directly competes with this oligomerization site. Interaction of the Rad52 C-terminal tail may favor proper Rad51 filament formation and also protect against the action of the Srs2 helicase (in green), which is known to interact with ssDNA and to translocate from the 3' to 5' end.

structural feature is conserved across all eight BRC motifs (Supplementary Fig. 8C). Remarkably, the RAD51 surface contacted by this helix is distinct from the surface that interacts with the extensions of the F316xxA motif of yeast Rad52. The C-terminal motif of Rad52 is not present in BRCA2 (Supplementary Fig. 8D), making fungi Rad52 unique in their ability to bind Rad51 through a single extended binding site with multiple contacts.

Still, as with Rad52, BRCA2 has also been implicated in RAD51 filament stabilization as recently shown[75]. This could likely be through its TR2 domain at the C-terminus, outside of the BRC repeats, which binds another surface of RAD51 (Supplementary Fig. 8E)[76]. Additionally, in plants, BRCA2 can antagonize FIGL1, a negative regulator of RAD51 filament formation[77], thereby extending the protective function against helicases action to both proteins.

Although their Rad51 binding modes have diverged, Rad52 and BRCA2 share conserved features. In both proteins, the short FxxA motif, is part of a long disordered region. This disordered nature, combined with the presence of conserved aromatic residues, is also observed in other specialized molecular chaperones, such as histone chaperones[78–80]. Interactions mediated by long disordered segments that fold upon binding to a globular partner allow coverage of a broad interaction surface while maintaining moderate affinity, due to a balance between entropic and enthalpic contributions, favoring dissociation of the chaperone after formation of the complex with other partners[79,81]. Intrinsic disorder may also favor the formation of condensates and repair foci[82].

Another feature likely important for Rad51 filament assembly is the presence of multiple Rad51-binding motifs in close proximity. In Rad52, this proximity arises from its ability to oligomerize through its N-terminal domain[25]. In contrast, BRCA2, which lacks this annealase domain, contains multiple BRC repeats, the number of which varies between species[83]. We propose that this multisite organization, together with the intrinsically disordered nature of Rad51 mediators Rad52

and BRCA2, could be critical for their chaperone function in Rad51 filament assembly.

## Methods

### Recombinant protein production and purification
The Rad52 (295–394), Rad52 (295–408) and Rad52 (206–471) fragments WT or with mutations F316A, F337A or Y376A were cloned in frame with 6his-GST in pETM30 plasmid. Peptides were purified from *E. coli* BL21(DE3) cells. Cells were grown in 2 L of minimum media M9 containing 15 μg/ml kanamycin and enriched with 0.5 g/L of $^{15}NH_4Cl$ and/or 2 g/L of $^{13}C$-glucose until OD reached 0.9. Protein expression was induced by adding 1 mM IPTG followed by over-night incubation at 37 °C. Cells were pelleted by centrifugation and resuspended in the lysis buffer (20 mM $NaH_2PO_4$ pH 7.4, 0.5 mM EDTA, 1 mM DTT, 150 mM NaCl, 5 μg/ml Chymostatin, 5 μg/ml Pepstatin, 1x Complete EDTA free Protease Inhibitor Cocktail (Roche), 1 mM AEBSF) containing 0.1 % NP-40. Cells were lysed by addition of 1.25 mg/ml Lysozyme for 45 min at 4 °C, followed by sonication at 4 °C. The lysate was clarified by centrifugation at 40 000 rpm for 45 min and incubated over-night with 2 ml of GSH-sepharose 4B resin (Cytiva) at 4 °C. The resin was washed with 100 ml of lysis buffer containing 150 mM KCl. Elution was then realized with 2.5 ml of lysis buffer containing 20 mM L-Gluthatione (Sigma). The 6his-GST tag was cleaved by adding the protease 6his-TEV to the eluate (1 mg TEV per 20 mg protein) and incubated overnight at 4 °C. 6his-TEV and 6his-GST were removed from the supernatant by incubating the eluate with Ni Sepharose High Performance resin (Cytiva) in lysis buffer without DTT, supplemented with 500 mM NaCl and 20 mM Imidazole. The flow-through containing the Rad52 peptide was then loaded in a PD10 column (Cytiva). The peptide was eluted with buffer G (20 mM NaH2PO4 ph 7.4, 0.5 mM EDTA, 1 mM DTT, 150 mM NaCl), then concentrated with Amicon Ultra-4, 3KDa filter concentrator (Merck). Rad52 (295–394), Rad52 (295–408) and Rad52 (206–471) concentrations were determined using an extinction coefficient of 4470 mole/L/cm at 280 nm.

Rad51 was overexpressed in *E. coli* BL21 (DE3) pLysS cells transformed with the pEZ5139 plasmid (provided by S. Kowalczykowski) and then purified as described previously[84].

### Size-exclusion chromatography (SEC)
Rad51, Rad52 fragments and complexes at 23 μM were incubated overnight at 4 °C and then injected into a Superdex 200 increase 3.2/300 (Cytiva) for separation by size-exclusion chromatography previously equilibrated with the buffer (20 mM Tris-HCl pH 8, 100 mM NaCl, 1 mM DTT). The different fractions were analyzed on 4-15 % Mini-PROTEAN TGX® Precast Gels (Bio-Rad) with 2.5 mM Tris, 20 mM Glycine, 0.01 % SDS running buffer.

### Nuclear Magnetic Resonance (NMR)
NMR experiments were performed on Bruker DRX-600, DRX-700 or Neo-700MHz spectrometers equipped with cryo-probes. All NMR data were processed using Topspin 3.5 (Bruker) and analyzed using Sparky 3.114 (T.D. Goddard and D.G. Kneller, UCSF). Samples were prepared in 3 mm NMR tubes, in a 20 mM Phosphate buffer solution pH 7.4, NaCl 150 mM, DTT 1 mM, EDTA 0.5 mM protease inhibitor cocktail 1x (Roche), 5 % $D_2O$, 0.1 % $NaN_3$, 0.1 mM DSS. Heteronuclear Multiple Quantum Correlation (sofast-HMQC) or best-HSQC spectra were all recorded at 283°K. The protein concentrations ranged from 10 μM and 200 μM. For backbone resonance assignments, 3D spectra were collected at 283°K using standard pulse sequences provided by Bruker for HSQC, sofast-HMQC, HNCA, HN(CO)CA, HNCACB, CBCA(CO)NH, HN(CA)CO, HNCO, and HNCANH experiments. Proton chemical shifts (in ppm) were referenced relative to internal DSS and $^{15}N$ and $^{13}C$ references were set indirectly relative to DSS using frequency ratios[85]. Chemical shift indexes were calculated according to the sequence-specific random coil chemical shifts[86,87]. For HSQC, sofast-HMQC

spectra, the number of scans varied between 64 and 256, spectral width in the amide proton dimension was between 14 and 16 ppm with a frequency offset of 4.70ppm (acquisition time between ~ 0.056 s and ~0.067 s). In the amide nitrogen dimension, spectral width was set between 22 and 36 ppm with a frequency offset of 118 ppm.

## Small angle xray scattering (SAXS)

SAXS data were collected at the SWING beamline on a EigerX 4 M detector using the standard beamline setup in SEC mode[88]. Samples were injected into a Superdex 5/150 GL (Cytivia) column coupled to a high-performance liquid chromatography system, in front of the SAXS data collection capillary. The initial data processing steps including masking and azimuthal averaging were performed using the program FOXTROT[89] and completed using US-SOMO[90]. The final buffer subtracted and averaged SAXS profiles were analyzed using BioXTAS RAW software package[91,92]. To model the structures and improve the AlphaFold2 models, the program Dadimodo[93] (https://dadimodo.synchrotron-soleil.fr) that refines multidomain protein structures against experimental SAXS data was used (see Supplementary Table 3 for more information).

## Structural modeling

Sequences of *S. cerevisiae* Rad51 (Uniprot ID: P25454) and Rad52 (NCBI ID: AHY76431.1) were used as input of Mmseqs2 homology search program[94] with three iterations to generate a multiple sequence alignment (MSA) against the uniref30_2103 database[95]. Note that the Rad52 amino acids are numbered from the first AUG codon in *RAD52* mRNA[58]. The resulting alignments were filtered using hhfilter[96] using parameters ('id'=100, 'qid'=25, 'cov'=50) and the taxonomy assigned to every sequence keeping only one sequence per species. Full-length sequences in the alignments were then retrieved and the sequences were realigned using MAFFT[97] with the default FFT-NS-2 protocol. The size of the individual MSAs was then restricted following two sets of delimitations Rad51 (1–400) vs Rad52 (295–408) and Rad51 (77–400) vs Rad52 (310–394) to enhance the sensitivity of AlphaFold2 detection for interactions involving intrinsically disordered regions[98]. The delimitated MSAs were concatenated so that when homologs of different subunits belonged to the same species, they were aligned in a paired manner, otherwise they were left unpaired[99]. Concatenated MSAs were used as inputs to generate 5 structural models for each of the delimitations sets using AlphaFold v2.3 through a local version of the ColabFold interface v1.5.2[100] trained on the multimer dataset[101]. Four scores were provided by AlphaFold2 to rate the quality of the models, the pLDDT, the pTMscore, the ipTMscore and the model confidence score (weighted combination of pTM and ipTM scores with a 20:80 ratio). The scores of the best longer model of the Rad51 (1–400) vs Rad52 (295–408) are (pLDDT = 72.2, pTM = 0.72, ipTM = 0.69, confidence score=0.70) and of the shorter Rad51 (77–400) vs Rad52 (310–394) delimitations are (pLDDT = 84.9, pTM = 0.80, ipTM = 0.85, confidence score = 0.84). The models ranked first for each complex were relaxed using OpenMM engine[102] and AMBER force field. The model of the complex Rad51 (77–400) vs Rad52 (310–394) was deposited on the ModelArchive database (https://modelarchive.org/) with the entry code ma-7m6mt. Sequence conservation in Rad52 and Rad51 was represented using the weblogo server (https://weblogo.berkeley.edu/logo.cgi) the conservation grades were calculated using the rate4site algorithm[103]. For Rad52, due to strong sequence divergence, two alignments were used, either restricted to Saccharomycetaceae species containing 44 homologs or containing 250 representative sequences of all fungal species with redundancy filtered at 80% sequence identity. For Rad51, the alignments used contained 100 sequences of fungal species filtered at 85% sequence identity. The Molecular graphics and analyses were performed with UCSF ChimeraX[104]. The disorder probability was calculated with the IUPred2A software (https://iupred2a.elte.hu)[105].

## S. cerevisiae strains

The strains used in this study are listed in Supplementary Data 1. Most experiments were performed in the FF18733 background. The 3His-6FLAG epitope was cloned in frame with *RAD52* into centromeric plasmids by SLIC[106]. Centromeric and replicative 2 μ plasmids carrying mutations in the *RAD52* or *RAD51* genes were transformed into yeast cells by one-step transformation[107]. Amino acids are numbered from the first AUG codon in *RAD52* mRNA[58]. All deletion mutants were constructed by the one-step gene disruption method[108]. *RAD51* and *RAD52* mutations were inserted into the yeast genome using the pop-in-pop-out technique. Insertions of the 3His-6FLAG epitope in phase with *RAD52* were constructed as previously described[109].

## Directed mutagenesis

Single mutations were introduced into the *RAD52* or *RAD51* genes cloned on plasmids (Supplementary Data 1) using a one-step PCR-based site-directed mutagenesis method adapted from reference[110]. Mutagenic primers carrying the desired mutations were designed as complementary oligonucleotides. Following PCR amplification, the reaction mixtures were treated with DpnI to digest the methylated parental plasmid DNA. The resulting PCR products were then transformed into *E. coli*, and plasmid DNA was extracted from two independent transformants. Plasmids were subsequently verified by whole plasmid sequencing (Eurofins Genomics).

## Co-immunoprecipitation

Yeast cells were grown in rich medium (YPD) to a concentration of $2.5 \times 10^6$ cells/ml. Cells were harvested and washed twice with PBS. Extracts were prepared as described previously[111]. Rad52-Rad51 were co-immunoprecipitated as described in ref. 41. Whole-cell extracts (1 mg) were incubated (4 °C for 1 h) with 0.5 μl of anti-Rad51 polyclonal antibody (produced in rabbit by Eurogentec) for Rad51 immunoprecipitation. Then, 50 μl of Dynabeads coupled to Protein A (Invitrogen) were added, and the incubation continued for another hour. Immunoprecipitates were washed twice with 1 ml of lysis buffer (Hepes-KOH pH 7.5 50 mM, NaCl 140 mM, EDTA 1 mM, Glycérol 10 %, NP40 0.5 %, AEBSF 1 mM, Protease inhibitor cocktail Complete 1x, Roche) and resuspended in 30 μl of Laemmli buffer (1x). The eluted proteins were analyzed by western blotting. Proteins were separated on Tris-glycine 4-15% (Biorad) and transferred to PVDF membranes. Proteins were detected with mouse anti-FLAG monoclonal (Sigma, 1/10,000) or rabbit anti-Rad51 polyclonal (1/2000) antibodies. Blots were then incubated with monoclonal goat anti-mouse Alexa800 (1/10,000, Lifetech) or goat anti-rabbit IR700 or IR800 secondary antibodies (1/10,000, Advansta). Protein-antibody complexes were visualized using the IBright system (Thermofisher). The presence of Rad51 in the immunoprecipitated fractions could not be detected to validate the efficiency of the immunoprecipitation because it migrates at the same level as the anti-Rad51 IgG. However, the absence of Rad52 in the *rad51Δ* immunoprecipitates confirmed that the detected Rad52-FLAG signals were related to the Rad52-Rad51 interaction. Rabbit Anti-tubuline and anti-Pgk1 antibodies (1/2000) were provided by Abcam.

## Irradiation and cell survival assay

Cells in the exponential growth phase were used for irradiation with γ-rays. γ-irradiation was performed using a GSR D1 irradiator (Gamma-Service Medical GmbH). It is a self-shielded irradiator with four sources of $^{137}$Cesium. The total activity was 180.28 TBq in March 2014. As yeast cells resuspended in 1 mL of sterile $H_2O$ were irradiated in 1.5 ml plastic microtubes, dosimetry was performed using plastic microtubes with a cylindrical ionizing chamber 31010 (PTW, Freiburg, Germany) following the American Association of Physicists in Medicine recommendations[112]. This ionizing chamber has a cavity of 0.125 cm³ calibrated with $^{137}$Cesium in air kerma free in air and the reference number of our facility is 210382401. The polarity and the ion

recombination were measured for this [137]Cesium source. Each measurement was corrected with the KTP factor to take into account the variations in temperature and atmospheric pressure. Yeast cells were irradiated at 100, 200, 400 and 600 Gy (single doses) and with a 12 Gy/min dose rate that takes the radioactive decrease into account.

Cells were plated at the appropriate dilution on YPD to measure the survival rate. The mean percentage from at least two independent experiments is presented.

## Survival following DNA DSB formation

Cells were grown overnight in liquid culture medium containing lactate (YPL) instead of glucose before plating. Survival following HO-induced DNA DSB was measured as the number of cells growing on galactose-containing medium divided by the number of colonies growing on glucose-containing medium. The presented results are the mean of at least three independent experiments.

## ChIP experiments and quantitative PCR analyses

Cells were grown in YPD until late exponential phase. After inoculation in 400 ml of YPL, cultures were grown to a concentration of $5 \times 10^6$ cells/ml. A 50-ml fraction was removed at the 0-h time-point, and then galactose was added to a final concentration of 2%. Incubation was continued and 50-ml fractions were removed at 4 h. Cells were fixed in 1% formaldehyde, which was then neutralized with 125 mM glycine. Cells were centrifuged and washed with TBS buffer (20 mM Tris pH 8, 150 mM NaCl). Cell pellets were then frozen at −20 °C. ChIP and quantitative PCR analyses was carried out as previously described[41].

## Y2H analysis

CTY10-5d strain and derivatives were transformed with both pBTM116 and pACT2-derived plasmids containing the genes of interest with the Gal4AD sequence or the LexADBD sequence at the N-terminus. β-galactosidase activity was measured with the semi-quantitative yeast β-galactosidase activity kit (Thermo Scientific) on individual co-transformant colonies. Expression of protein fusions often results in slow growth, which is probably related to a toxicity of overexpression of the protein fusions and is at the origin of the variability in β-galactosidase activity we observed. The activity measured is only semi-quantitative because it is related to the fitness of the colony tested. Therefore, the results can only be read qualitatively as a strong interaction, a weak interaction or no interaction at all, at least within the limit of detection of the Y2H assay. We found that overexpressing Rad51 is highly toxic, resulting in a very low transformation frequency. However, introducing a *rad51-E221K* mutation at the Rad51-Rad51 interface suppresses the toxicity of Rad51 overexpression without disrupting Rad51-Rad51 interaction (Fig. 7A). Accordingly, this mutation was systematically used in conjunction with those tested for their impact on Rad51-Rad51 interaction. Plasmids bearing *RAD51* mutated alleles were always transformed in a CTY10-5d derivative bearing a complete deletion of *RAD51*.

## Microscopy

Yeast cells were grown in YPD overnight to early log-phase. To induce a single DSB at the I-SceI cut site, cells were grown overnight in YP 3% raffinose to early log phase. In the morning, the cultures were normalized at 0.4 OD$_{600nm}$/ml in YP-3% Raffinose, after 2 h, galactose was added directly to the culture at a 2% final concentration. Before microscopy, cells were rinsed with complete synthetic medium (CSM 3% Raffinose) and were placed on a 1.4% agarose patch for microscopy. For all fluorescent images, images were acquired in three dimensions with a z-step of 200 nm. The images shown are a maximum intensity projection of the z-stack images. Images were acquired on an inverted wide-field microscopy (Nikon TE2000) equipped with a Å-100/1.4 numerical aperture (NA) immersion objective and a sCMOS camera

(BSI, Photometrics); the pixel size is 6.5 μm. A spectra X light engine lamp (Lumenor) was used to illuminate the samples.

## Quantification of Rad51 structures

**Image processing.** Yeast nuclei were cropped to 50 × 50 pixels size images and deconvolved using SVI Huygens with the Classical Maximum Likelihood Estimation algorithm at 25 iterations and a quality change stopping criterion of 0.01. All crops are then processed by a Fiji macro to extract filament shape and intensity features as well as eventual foci information. All segmentations, filaments and foci, are performed on deconvolved images and based on generalized Laplacian of Gaussians filters[113]. For the filaments, the skeleton of the resulting masks is then processed (pruned and elongated) to better fit the shape of the filament. The shapes and complexity of segmented filaments are described through the Analyze Skeleton plugin (https://imagej.net/plugins/analyze-skeleton/). To ensure reproducibility, a single parameter corresponding to the average width of filaments (here 2 pixels, 130 nm) has to be set by the user. Intensity quantifications are done on the raw images with subtracted yeast nuclear background (removal of the median gray level value of the yeast nucleus without the filament). On top of the presence of filaments, three features are kept: the length of the filaments, their total intensities and the eventual foci intensities. Filament lengths shown in Fig. 6 and 7 correspond to simple filaments (unique unbranched skeleton).

**Statistical analysis.** For descriptive statistics, each crop is classified along four categories according to the presence and complexity of filaments and foci: Simple (defined as unique skeleton without branches), Complex (multiple or branched skeletons), Foci, and Nothing. As length measurement is subject to overestimation for complex filaments, they were not included in the quantification presented in Fig. 6 and 7. Also, in the case of several Foci, only the brightest was included. Logistic regression with a binomial distribution of the dependent variable is performed to test for association between pairs of mutants and the different image features. Three models are defined:

condition ~ 1 + hasFilament + batch
condition ~ 1 + filamentLength + filamentIntensity + batch
condition ~ 1 + spotIntensity + batch

A total of 52 hypotheses are thus tested to study the impact of Rad51-Rad52 binding and 17 more for Rad51-T161R, leading to an increased risk of false rejection. To control this risk, all the p-values from these multiple significance tests are corrected with the false discovery rate[114].

## Reporting summary

Further information on research design is available in the Nature Portfolio Reporting Summary linked to this article.

## Data availability

Chemical shifts of Rad52 C-terminal segment -206-471 is available on the BMRB under the ID BMRB 53001. The structural model of the complex between the globular domain of *S. cerevisiae* Rad51 (77–400) and the disordered C-terminus of Rad52 (310–394) is available in ModelArchive (modelarchive.org) with the accession code ma-7m6mt. [https://www.modelarchive.org/doi/10.5452/ma-7m6mt] Microscopy data shown in Fig. 6 and 7 as well as the complete Source Data file are available in Zenodo at https://zenodo.org/records/15149086. Source data are provided with this paper.

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

## Acknowledgements

This paper is dedicated to the memory of Dr. Francis Fabre. Francis passed away on December 1st 2024 from leukemia. Francis made a tremendous contribution to the field of DNA repair. He was also a fantastic mentor and will be remembered as a very generous person and a very good friend. We thank S. Kowalczykowski for the gift of the pEZ5139 plasmid. We thank the BIOI2 platform for making the ColabFold pipeline easily accessible at the I2BC. This work was supported by grants from EDF (F.O., E.C.), CEA Radiobiology call (E.C.), Tandem Call CEA-PIC3i Curie in Radiobiologie (A.T., E.C.), INCA (2016-1-PL BIO-03-CEA-1, F.O.), ANR (ANR-20-CE18-0038; ANR-21-CE11-0027; ANR-21-CE44-0009-01; ANR-23-CE11-0023 (A.T., F.O., E.C.), the program labeled by the ARC foundation 2016 (PGA1*20160203953, F.O.), Labex DEEP (ANR-11-LABEX-0044 DEEP and ANR-10IDEX-0001-02 PSL, A.T.) and by French infrastructures, the Synchrotron Soleil (20191119; 20210745), the French Infrastructure for Integrated Structural Biology (FRISBI) ANR-10-INBS-0005, the IR INFRANALYTICS FR2054 and the GIS IBISA. We also thank the PICT-IBiSA@Orsay and PICT-IBiSA@Pasteur Imaging Facilities of the Institut Curie, a member of the France Bioimaging National Infrastructure (ANR-10-INBS-04).

## Author contributions

E.M., F.L., E.L., M.R., M.A., N.L., E.N., L.M., J.D., C.B., A. Thureau., D.B. and X.V. performed the experimental studies. M. G. performed the microscopy statistical analyses. R.G. performed structural models. A. Taddei supervised the work and wrote the manuscript. F.O. and E.C. performed the experimental studies, supervised the work and wrote the manuscript.

## Competing interests

The authors declare no competing interests.

## Additional information

[1]Université Paris Cité, Inserm, CEA, Stabilité Génétique Cellules Souches et Radiations, LRGM/iRCM/IBFJ, F–92260 Fontenay-aux-Roses, France. [2]Université
Paris-Saclay, Inserm, CEA, Stabilité Génétique Cellules Souches et Radiations, LRGM/iRCM/IBFJ, F–92260 Fontenay-aux-Roses, France. [3]Nuclear Dynamics,
CNRS UMR 3664, Institut Curie, PSL Research University, Sorbonne Université, F–75005 Paris, France. [4]Institut Joliot, Commissariat à l'énergie Atomique
(CEA), Direction de la Recherche Fondamentale (DRF), F–91191 Gif-sur-Yvette, France. [5]Institute for Integrative Biology of the Cell (I2BC), CEA, CNRS, Univ.
Paris-Sud, Université Paris-Saclay, F–91198 Gif-sur-Yvette, cedex, France. [6]Université Paris Cité, Inserm, CEA, Stabilité Génétique Cellules Souches et
Radiations, CIGEx/iRCM/IBFJ, F–92260 Fontenay-aux-Roses, France. [7]Université Paris-Saclay, Inserm, CEA, Stabilité Génétique Cellules Souches et Radia-
tions, CIGEx/iRCM/IBFJ, F–92260 Fontenay-aux-Roses, France. [8]Synchrotron SOLEIL, HelioBio group, l'Orme des Merisiers, Départementale 128, F–91190
Saint-Aubin, France. [9]These authors jointly supervised this work: Angela Taddei, Françoise Ochsenbein, Eric Coïc. ✉e-mail: Angela.Taddei@curie.fr;
Francoise.OCHSENBEIN@cea.fr; eric.coic@cea.fr

