## [Transparent Peer Review file · Nature Communications]

A large C-terminal Rad52 segment acts as a chaperone to Form and Stabilize Rad51 Filaments

Corresponding Author: Dr Eric Coïc

Version 0:

Reviewer comments:

Reviewer #1

(Remarks to the Author)

In Ma et al. The authors deliver an excellent and timely paper discussing the ability of Rad52 to act as a molecular chaperone to Rad51. The experiments are technically well done, and the findings support the primary conclusions of the paper. The major conclusions are that a disordered surface of Rad52 interacts with the oligomerization of Rad51 to prevent self-association and promote loading on ssDNA. The authors also find that this region protects Rad51 filament disassembly by the motor protein Srs2. Although, the mechanism of this is less clear (See below). This excellent study represents a significant conceptual advance in our understanding of Rad51 filament formation. The one limitation of the study is a more defined relationship between the BRCA2 activity from higher eukaryotes and the activity of Rad52 in yeast if there were some relationship between the sequences identified that went beyond amino acid alignment, that would be cool. However, it is not necessary and in my opinion this manuscript is suitable for publication with some revision of the text. I have listed some major and minor comments below.

Major:

Rad52 and Brca2 are functional homologs for Rad51 loading only. BRCA2 lacks and annealase domain which is vital for function. This distinction is important.

This is an equilibrium problem. The evidence does not support direct protection of Rad51 by Rad52 on ssDNA. All it shows is that an equilibrium is shifted to maintain filament formation. I understand the EM images from (Ma et al. 2018) suggest direct protection. However, without a real-time observation of Rad51 filament protection, I don't think the conclusion of direct protection can be made. Similar arguments were made for Rad55/57 and Srs2 (Liu et al. 2011), and this was later found to be an EM artifact when Rad55/57 loading of Rad51 was directly visualized in real time (Roy et al. 2021). I think maybe tempering the language is appropriate here.

How well conserved are these regions with BRCA2? There is functional redundancy between the proteins, but I am wondering how that manifests on the protein level. Are there any other properties of these regions, outside of sequence, that might drive the interaction in both context? For example, are there flexibility properties that are similar but have different amino acids.

Minor:

Previously, we showed that deleting the Srs2 translocase, which efficiently displaces Rad51 from ssDNA,

Supplementary Figure 5: Please add a citation or permission for the data from a previous publication.

Reviewer #2

(Remarks to the Author)

Yeast Rad52 and its functional homolog in human, the tumour suppressor BRCA2, promote the actions of the strand exchange protein Rad51/RAD51 to mediate homologous recombination reactions. This interplay is essential for chromosome stability, remains incompletely understood, and studies in this area are topical and of great interest to the field. The 'mediator' role of Rad52 and BRCA2 comprises (1) the recruitment of the Rad51/RAD51 recombinase, and (2) enabling productive Rad51/RAD51 nucleoprotein filament formation on single-stranded DNA. Interactions between multiple BRCA2 'BRC repeats' and RAD51 have been characterized in detail; interestingly BRC motifs dissociate RAD51 oligomers in solution by mimicking the RAD51 protomer-protomer interface, thereby competing with RAD51 oligomerization (ref. 14 in the current MS), while BRCA2 and Rad52 also stabilize RAD51/Rad51 oligomers on DNA (nucleoprotein filaments). Interactions like those between the BRCs and RAD51 occur between motifs in the C-terminal domain of Rad52 and Rad51 in yeast (e.g., ref. 53). A recent cryo-EM study has shown that Rad52 functions as a homodecamer, where the structured N-terminal domain of each subunit forms one unit of a ring structure. The respective C-terminal half of each subunit, which contain the binding sites for Rad51 as well as for single-stranded DNA binding protein RPA, remained disordered and structurally intractable (ref. 25).

The current study "Rad52 Acts as an Assembly Chaperone to Form and Stabilize Rad51 Filaments Through a Large C-Terminus 85-Residue Segment" by Emilie Ma and co-workers addresses the details of the Rad52 C-terminal interactions with Rad51 in *Saccharomyces cerevisiae*, extending on a previous report (ref. 39). The authors demonstrate a three-pronged (as opposed to a previously suggested two-pronged) interaction between the Rad52 C-terminus and Rad51 and provide a comprehensive mutational analysis to deduce the relative importance of each touchpoint for Rad52 mediator function. There are a few specific points which could be improved, for example including western blots to accompany the quantifications of the Rad52-Rad51 co-IP data (e.g., in Supplementary Figure S3G). It is also not clear how the authors rationalize that rad51-T161R mutant shows complete loss of Rad52 binding yet only a mild MMS sensitivity phenotype (Fig. 7), whilst structure-guided point mutations in Rad52 that lose interaction with Rad51 resulted in MMS hypersensitivity (see conclusion on page 9, "Interestingly, mutants that exhibited a complete loss of interaction with Rad51, as assessed by co-immunoprecipitation (coIP), were sensitive to low doses of MMS (0.01%), highlighting the functional importance of this interaction."). Generally speaking, the data is of high quality, the protein modelling is appropriately presented and validated, and the conclusions are sound.

Having said that, there is a major concern that the structural and functional results in this study largely confirm existing data and the conceptual advance appears to be limited.

In their preceding 2018 study (ref. 39), the authors concentrated on a region within Rad52 spanning amino acid residues 316 to 379, which contains two previously established Rad52-binding motifs (FVTA) at position 316-319 and (YEKF) at position 376-379 (defined by the Sung and Shibata/Kurumizaka groups in refs. 51 and 53). In the present MS, it is roughly the region demarcated by these known Rad52 motifs that the authors re-label as an 85 amino-acid Rad51 interaction domain (spanning residues 310-395). This domain is found to contain a third motif (residues 337-367) supporting Rad51 binding. An extensive mutational analysis paired with DNA damage-sensitivity spot assays, co-immunoprecipitation of Rad52 and Rad51, and recombination-dependent DNA repair genetic assays is largely consistent with the notion that, as predicted by AlphaFold, there are three motifs (the two previously defined FVTA and YEKF motifs, as well as a related motif starting at F337 of Rad52) mediating three-pronged Rad51 monomer interactions and supporting Rad51 recruitment and nucleoprotein filament formation on single-stranded DNA.

While the key literature is cited, the findings within this MS are not well integrated with the existing literature. For the most part, the results presented are congruent with the current consensus on Rad52's mediator role. For example, section 1 of Results "A large segment of 85 residues in the center of the fully disordered Rad52 C-terminal domain interacts with Rad51" and Fig 1 overlap largely with findings in refs. 51 and 53. However, refs. 51 and 53 are not cited or discussed in this section, which makes it difficult for the reader to place the MS in its proper context and identify where new insight arises. A similar criticism applies to the subsequent section and Fig. 2, which also appears largely redundant with the literature (Rad52 binding to Rad51 promoting Rad51 oligomer dissociation, e.g. ref. 53). With respect to the main conclusion and working model in Fig. 8, similar models have been published previously for BRCA2 as well as Rad52, and while some of these previous publications are cited, this is not clearly discussed in the Discussion section to identify new advances specific to the current MS. A discussion of the model for Rad52-Rad51 interactions/Rad51-chaperoning presented recently by Deveryshetty and co-workers (ref. 25) is missing. The Deveryshetty model (Fig. 8 in ref. 25) is very similar to the one presented herein (Fig. 8 in the current MS). Moreover, the authors do not relate the ideas illustrated in their model (Fig. 8) to the suggestions of Deveryshetty and co-workers who, besides C-terminal Rad52-Rad51, also show N-terminal Rad52-Rad51 interactions that are important for Rad52's mediator function.

Kagawa (2013) and co-workers (ref. 53) have previously noted the striking similarity between the Rad51-binding motifs in Rad52 and the RAD51-binding BRC motifs in BRCA2, and they have published a homology model for the role of the Rad52 and BRCA2 motifs in sequestering monomeric Rad51/RAD51 to mediate proper loading onto single-stranded DNA/nucleoprotein-filament nucleation and growth (see Fig. 7 in ref. 53, similar to the iteration in Supplementary Fig. 4 in the current MS). Based on their structural data, Deveryshetty and co-workers (ref. 25) envisage a Rad51 chaperone model, where the C-terminal tails of Rad52 extending from the Rad52 homodecameric ring (formed by the Rad52 N-terminal domains) sequester Rad51 molecules and deliver them for assembly onto DNA (ref. 25).

An interesting point of novelty within the current MS is the three-pronged (rather than two-pronged) nature of interaction between Rad52 and Rad51. In addition, by structural analysis/modelling, the authors succeed in providing an improved view of Rad52-Rad51 interactions. The work is solid, interesting, and should be published; however, it does not provide the kind

of conceptual advance over what we know about Rad52/BRCA2-Rad51/RAD51 mediator function that would make the MS a strong candidate for publication in Nature Communications.

Reviewer #3

(Remarks to the Author)

This paper studies the structural basis for the interactions between Rad51 and Rad52, and the role of these interactions in RAD51 filament formation and filament stability. The authors have identified the region spanning residues 310-395 in the C-terminal of Rad52 as the Rad51-interacting region using NMR. They have then generated a model of the Rad51-Rad52 interaction using the NMR data and combining it with SAXS data and AlphaFold modelling. Multiple sets of experiments, both in vitro and in vivo, have then been carried out to verify this model using site directed mutants. The authors conclude that the interaction consists of three motifs, two of which bind to the Rad51 multimerization surface to prevent oligomerization and facilitate loading of monomers. The third motif is seen to stabilise Rad51 filaments by preventing disassembly by Srs2. Based on their results, they then propose a mechanistic model for the loading of Rad51 onto ssDNA by Rad52.

The study is novel and rigorous, and the molecular mechanisms involving Rad51 are important for the fields of DNA replication and repair. My concerns are listed below, and there are some minor language corrections I have annotated in the file. I recommend that this paper be accepted once these concerns are satisfactorily resolved.

Concerns:

1) The results section titled 'Rad52 (310-394) folds upon Rad51 binding burying a large conserved surface' describes the interaction interface of Rad51-Rad52, but this description is poorly quantified in the text and not clearly visualised in Fig. 3. An analysis of the surface to include the residue-level interactions such as H-bonding, pi-stacking should be added, in addition to the data on surface area and buried residues. The figures show Rad51 as a surface, figures with the individual interacting residues as sticks with corresponding bonds would help with visualisation. Fig. 3A has some green dotted lines in it but no explanation is given in the caption.

2) A similar issue is present in the section titled 'Rad52 (310-394) competes with Rad51 oligomerization'. A further description of the binding pockets of the anchors and their interactions should be added, and the Fig. 4B panels are hard to visualise due to the surface map over the ribbon diagram.

3) Fig. 4C shows graphs with intensity comparisons of the spectra of the Rad52 mutants bound to Rad51 vs the WT. One concern while designing mutations in proteins is if the resulting mutants are folded and stable. In this case adding the HSQC spectra of the mutants overlaid with the WT in the supplementary material will be helpful.

4) In supplementary Fig. 3F it is extremely hard to distinguish between the 5 models, a colour scheme with better contrast between the models would be helpful.

Version 1:

Reviewer comments:

Reviewer #1

(Remarks to the Author)

The authors have addressed all of my concerns. Congratulations on this cool study.

Reviewer #2

(Remarks to the Author)

The revised version of the MS addresses most points raised. Further textual clarifications are strongly recommended as detailed below.

Point by point answer to the reviewers' comments:

Reviewer #2 (Remarks to the Author)

Yeast Rad52 and its functional homolog in human, the tumour suppressor BRCA2, promote the actions of the strand exchange protein Rad51/RAD51 to mediate homologous recombination reactions. This interplay is essential for chromosome stability, remains incompletely understood, and studies in this area are topical and of great interest to the field. The 'mediator' role of Rad52 and BRCA2 comprises (1) the recruitment of the Rad51/RAD51 recombinase, and (2) enabling productive Rad51/RAD51 nucleoprotein filament formation on single-stranded DNA. Interactions between multiple BRCA2 'BRC repeats' and RAD51 have been characterized in detail; interestingly BRC motifs dissociate RAD51 oligomers in solution by mimicking the RAD51 protomer-protomer interface, thereby competing with RAD51 oligomerization (ref. 14 in the current MS), while BRCA2 and Rad52 also stabilize RAD51/Rad51 oligomers on DNA (nucleoproteinfilaments). Interactions like those between the BRCs and RAD51 occur between motifs in the C-terminal domain of Rad52 and Rad51 in yeast (e.g., ref. 53). A recent cryo-EM study has shown that Rad52 functions as a homodecamer, where the structured N-terminal domain of each subunit forms one unit of a ring structure. The respective C-terminal half of each subunit, which

contain the binding sites for Rad51 as well as for single-stranded DNA binding protein RPA, remained disordered and structurally intractable (ref. 25).

The current study “Rad52 Acts as an Assembly Chaperone to Form and Stabilize Rad51 Filaments Through a Large C-Terminus 85-Residue Segment” by Emilie Ma and co-workers addresses the details of the Rad52 C-terminal interactions with Rad51 in *Saccharomyces cerevisiae*, extending on a previous report (ref. 39). The authors demonstrate a three-pronged (as opposed to a previously suggested two-pronged) interaction between the Rad52 C-terminus and Rad51 and provide a comprehensive mutational analysis to deduce the relative importance of each touchpoint for Rad52 mediator function.

We thank the reviewer for summarizing key background information on the topic and for highlighting previous publications, which we also cite in our manuscript. We would like to emphasize, however, that the present study goes significantly beyond our previous report (ref. 39). In that earlier work, we showed that Rad52–Rad51 interaction is not strictly required for Rad51 filament formation, we also observed that Rad52 mutants disrupting this interaction suppress filament toxicity in *srs2Δ* cells and produce filaments that are more sensitive to Srs2-mediated disassembly. These findings led us to propose that Rad52 also stabilizes Rad51 filaments, though the mechanism was unknown.

Here, we provide the first detailed structural description of the Rad52–Rad51 interface, revealing an extended 85-residue Rad52 segment using NMR, AlphaFold, and SAXS. We show that Rad52 binds a single Rad51 monomer in competition with Rad51–Rad51 oligomerization, supporting a model in which Rad52 acts as a chaperone to isolate monomers and load them onto ssDNA.

As the reviewer notes, we tested this model through a comprehensive mutational analysis. Importantly, we extended this approach to assess the role of each contact site not only in mediator activity, but also in Rad51 filament formation by live microscopy and competition with Srs2—an unexplored aspect until now.

1) There are a few specific points which could be improved, for example including western blots to accompany the quantifications of the Rad52–Rad51 co-IP data (e.g., in Supplementary Figure S3G).

We added the western blots in addition to the quantification in Supplementary Figure S3G and 3H of the revised manuscript (cited page 13 of the main text).

Referee response: The WBs are a valuable addition, although one might argue that showing immunodetection for IPed Rad51 rather than WCE Rad51 would be the appropriate control here.

2) It is also not clear how the authors rationalize that *rad51-T161R* mutant shows complete loss of Rad52 binding yet only a mild MMS sensitivity phenotype (Fig. 7), whilst structure-guided point mutations in Rad52 that lose interaction with Rad51 resulted in MMS hypersensitivity (see conclusion on page 9, “Interestingly, mutants that exhibited a complete loss of interaction with Rad51, as assessed by co-immunoprecipitation (coIP), were sensitive to low doses of MMS (0.01%), highlighting the functional importance of this interaction.”).

The sentence on page 9 cited by the reviewer refers to a general correlation between loss of Rad52–Rad51 interaction and MMS sensitivity, without yet detailing the variability among mutants. To clarify this point, we have specified page 9 in the revised manuscript that mutants losing Rad52–Rad51 interaction show MMS sensitivity to varying degrees.

As noted in the original text (page 13), “*rad52-F316A* and *F337A* (anchors 1 and 2) exhibit higher sensitivity to MMS (Figure 3D) compared to *rad52-Y376A* (anchor 3).” A similar gradient between mutants was also observed after γ -irradiation (page 13 of the original text). On page 18 we compared *rad51-T161R* mutant with *rad52-Y376A*, stating that they display similar MMS sensitivity: “...comparable to that of *rad52-Y376A* cells (Figure 7B).” To avoid any confusion, we have now explicitly added page 18 a reference to Figure 3D in the revised manuscript to clarify this comparison.

We would like to emphasize that while all three anchor mutations disrupt Rad52–Rad51 interaction (as shown by NMR and co-IP), their *in vivo* effects are more nuanced. As emphasized in the discussion, we stress the importance of combining *in vitro* and *in vivo* approaches to detect subtle phenotypes that may be influenced by additional cellular factors. The range of sensitivities observed supports the distinct functional contributions of each anchor to Rad51 filament dynamics.

Referee response: OK.

3) Generally speaking, the data is of high quality, the protein modelling is appropriately presented and validated, and the conclusions are sound. Having said that, there is a major concern that the structural and functional results in this study largely confirm existing data and the conceptual advance appears to be limited.

In their preceding 2018 study (ref. 39), the authors concentrated on a region within Rad52 spanning amino acid residues 316 to 379, which contains two previously established Rad52-binding motifs (FVTA) at position 316-319 and (YEKF) at position 376-379 (defined by the Sung and Shibata/Kurumizaka groups in refs. 51 and 53).

In the present MS, it is roughly the region demarcated by these known Rad52 motifs that the authors re-label as an 85 amino-acid Rad51 interaction domain (spanning residues 310-395). This domain is found to contain a third motif (residues 337-367) supporting Rad51 binding. An extensive mutational analysis paired with DNA damage-sensitivity spot assays, co-immunoprecipitation of Rad52 and Rad51, and recombination-dependent DNA repair genetic assays is largely consistent with the notion that, as predicted by AlphaFold, there are three motifs (the two previously defined FVTA and YEKF motifs, as well as a related motif starting at F337 of Rad52) mediating three-pronged Rad51 monomer interactions and supporting Rad51 recruitment and nucleoprotein filament formation on single-stranded DNA.

We thank the reviewer for acknowledging the quality of our data.

Contrary to what is stated by the reviewer, our 2018 study (ref. 39) focused solely on a narrow segment of Rad52 (residues 371–387), corresponding to the YEKF motif. In contrast, the present work defines and characterizes, for the first time, the entire Rad51-binding domain of Rad52 (residues 310–395), using a combination of NMR, AlphaFold modelling, SAXS, and extensive *in vivo* functional assays.

The two previously known motifs (FVTA and YEKF) were primarily characterized *in vitro* (ref. 51 and 53), with only limited *in vivo* analysis that did not account for Srs2-dependent effects. In this study, we identify and validate a third functional Rad51-binding motif, centered on F337, and demonstrate that Rad52 uses three distinct “anchors” to engage Rad51 monomers.

Notably, we show that anchors 1 and 2 compete with the Rad51–Rad51 interface, thereby directly influencing filament formation, while anchor 3 could play a specific role in protecting filaments from Srs2 disassembly, with less impact on filament assembly itself.

These findings go well beyond confirming previous data. They define a cohesive structural and functional framework for Rad52's mediator activity, clarify the differential contributions of each interaction site to filament dynamics, and establish a mechanistic link between Rad52 architecture and its dual role in Rad51 loading and filament stabilization.

4) While the key literature is cited, the findings within this MS are not well integrated with the existing literature. For the most part, the results presented are congruent with the current consensus on Rad52's mediator role. For example, section 1 of Results "A large segment of 85 residues in the center of the fully disordered Rad52 C-terminal domain interacts with Rad51" and Fig 1 overlap largely with findings in refs. 51 and 53. However, refs. 51 and 53 are not cited or discussed in this section, which makes it difficult for the reader to place the MS in its proper context and identify where new insight arises.

Refs. 51 (Krejci et al. 2002) and 53 (Kagawa et al. 2014) are cited in the Introduction, prior to the presentation of our results. To help readers better contextualize our findings in light of these previous studies, we have now added explicit citations to refs. 51 and 53 in line 3 of the first paragraph of the Results section (page 5).

However, we would like to emphasize that these studies did not provide structural data to analyze the Rad52–Rad51 binding mode, as we do in Figures 1 to 4. Furthermore, both refs. 51 and 53 did not consider the contribution of Srs2 activity and incorrectly concluded that Rad52–Rad51 interaction is absolutely required for Rad51 filament formation. In contrast, our results demonstrate that mutants affecting this interaction can still form Rad51 filaments, but these filaments are highly sensitive to Srs2-mediated disassembly. Again, the comprehensive approach presented in this manuscript—including structural characterization, *in vitro* binding assays, and *in vivo* phenotypic analysis—provides a more integrated and robust framework for understanding Rad52's mediator activity and its role in regulating Rad51 filament dynamics.

Referee response: On re-reading the revised version of the MS, I still feel that previous work (ref. 51, 53) is not appropriately credited with defining the FVTA and YEKF Rad51-binding sites which are confirmed herein. The way the FVTA motif is introduced on p.4 is rather cryptic ("The Rad51 polymerization motif FVTA is conserved in Rad52 C-terminus and essential to disrupt Rad51 oligomers.") and the sentence makes no reference to the literature or the fact that the FVTA site has been defined as a conserved Rad51-binding domain with a notable similarity to BRC domains (ref. 53). Similarly, the Sung lab has shown by sequence alignment and mutational analysis that a second Rad51-binding site exists within Rad52 (YEKF) and is conserved (ref. 51). This should be made explicitly clear in the Introduction, which will help the reader appreciate where the data presented in the current MS confirms previous findings and conclusion, and which additional insight is won herein.

Adding refs. 51 and 53 in the first para of Results (p.5; "Although the C-terminal region of Rad52 (...) mediates essential interactions with the recombinase Rad51 [refs. 51,53] (...).") is an important improvement, but I fear the significance of this work will be lost on the reader. I strongly recommend the authors make it clear that these previous papers have defined clearly demarcated motifs that form the downstream and upstream boundaries of the 85-residue region defined in the current MS and that these motifs are equivalent to what is described here as 'anchor 1' and 'anchor 3'. This will provide much better context and, I feel, takes nothing away from the discovery of 'anchor 3' or the structural and functional implications described in the current MS. The potential impression that specific anchor points (1 and 3) have not been previously defined at amino acid level should be avoided, in particular since historic Rad52 residue numbering in refs. 51 and 53 could make it difficult for some readers to spot the congruence between the earlier work and the current paper. Appropriate textual changes are required before publication.

5) A similar criticism applies to the subsequent section and Fig. 2, which also appears largely redundant with the literature (Rad52 binding to Rad51 promoting Rad51 oligomer dissociation, e.g. ref. 53).

While Rad52's role in promoting Rad51 oligomer dissociation has been previously shown (e.g., ref. 53), the data presented in Figure 2 provide additional structural insight through SAXS analysis. This approach allows us to determine the stoichiometry of the complex and to compare it across various mutants under identical experimental conditions. We have now added page 7, a reference to the relevant literature in this section of the manuscript, as follows: "These results are fully consistent with previous analyses of this complex by SEC (ref. 53)."

Referee response: OK.

6) With respect to the main conclusion and working model in Fig. 8, similar models have been published previously for BRCA2 as well as Rad52, and while some of these previous publications are cited, this is not clearly discussed in the Discussion section to identify new advances specific to the current MS

Regarding the similarities and differences between BRCA2 and Rad52, the revised manuscript now includes a more complete, dedicated paragraph at the end of the Discussion section, in response to Reviewer #1's comment. This paragraph discusses their respective roles in Rad51 filament stability and highlights that anchor 3, which contributes specifically to filament protection, is absent from BRCA2. This underscores the unique ability of Rad52 to bind Rad51 through an extended interaction segment comprising three distinct regions, each exerting a subtle but specific impact on Rad52's mediator activity.

Referee response: OK.

7) A discussion of the model for Rad52-Rad51 interactions/Rad51-chaperoning presented recently by Deveryshetty and co-workers (ref. 25) is missing. The Deveryshetty model (Fig. 8 in ref. 25) is very similar to the one presented herein (Fig. 8 in the current MS). Moreover, the authors do not relate the ideas illustrated in their model (Fig. 8) to the suggestions of Deveryshetty and co-workers who, besides C-terminal Rad52-Rad51, also show N-terminal Rad52-Rad51 interactions that are important for Rad52's mediator function.

We agree that a comparative discussion of our model with the one proposed by Deveryshetty et al. (ref. 25) is a valuable addition to the manuscript. We have therefore added, a sentence in the Discussion page 21 section that refers directly to this study: "This oligomerization ability is conferred by its N-terminal annealase domain, whose structure was recently solved (ref. 25) (Supplementary Figure 1A). In that study, the authors proposed that the ssDNA-binding property of this domain, together with the association of Rad51 with the Rad52 C-terminal domain, could facilitate the transfer of Rad51 onto single-stranded DNA.". Ref. 25 is also now cited at the end of the Discussion about the multisite organization of Rad52 and BRCA2 mediators. Also we gave more details about the model in the legend of Figure 8.

Importantly, our model differs substantially from that of Deveryshetty et al., both in focus and in the nature of the supporting data. Their study primarily addresses the structural characterization of the N-terminal annealase domain and includes the C-terminal region in their model, but without experimental validation. In a complementary manner, our study focuses on the C-terminal domain of Rad52, and we integrate relevant insights from ref. 25 into our model. However, we do not attempt to depict interactions between the annealase domain and DNA, as this falls outside the scope of our experimental data.

Referee response: OK.

8) Kagawa (2013) and co-workers (ref. 53) have previously noted the striking similarity between the Rad51-binding motifs in Rad52 and the RAD51-binding BRC motifs in BRCA2, and they have published a homology model for the role of the Rad52 and BRCA2 motifs in sequestering monomeric Rad51/RAD51 to mediate proper loading onto single-stranded DNA/nucleoprotein-filament nucleation and growth (see Fig. 7 in ref. 53, similar to the iteration in Supplementary Fig. 4 in the current MS).

Indeed, the very interesting article published by Kagawa and co-workers (ref. 53) identified a clear similarity between the FxxA motifs of Rad52 and BRCA2. Figure 7 of that study presents the structure of Rad51 in complex with BRC4 by Pellegrini et al (ref 14 in our manuscript), and a sequence alignment of Rad52 homologues and, based on these information, this paper proposes that the Y376EKF motif in Rad52 could bind Rad51 similarly to F1524 in the BRC4 motif of BRCA2. However, this homology-based model is incorrect. Our structural analysis demonstrates that Y376 binds to a completely different site on Rad51 compared to F1524. To clarify this point, we have added Supplementary Figure 8, which directly compares the binding modes. Our study thus resolves a key inaccurate hypothesis from ref. 53, which arose due to the absence of reliable structural data in that earlier work.

Referee response: The conceptual point made by Kagawa et al. should be acknowledged in this context, with citation of ref. 53, regardless of whether their working hypothesis in the 2013 paper faithfully reflects the detailed protein interaction or not. I can't see where this is acknowledged, but if not done already, the insightful prediction by Kagawa et al. should be mentioned.

9) Based on their structural data, Deveryshetty and co-workers (ref. 25) envisage a Rad51 chaperone model, where the C-terminal tails of Rad52 extending from the Rad52 homodecameric ring (formed by the Rad52 N-terminal domains) sequester Rad51 molecules and deliver them for assembly onto DNA (ref. 25).

The model proposed by Deveryshetty et al. (ref. 25) suggests that each C-terminal tail of Rad52 binds two Rad51 molecules—an assumption not supported by experimental data. Furthermore, their study does not propose a chaperone model per se, but rather a scaffolding model in which each C-terminal region recruits multiple Rad51 molecules to generate a sufficient local concentration for filament nucleation. Notably, the concept of a Rad51 chaperone is never explicitly mentioned in the text of their paper.

Referee response: Semantics aside, the Deveryshetty model is conceptually very similar to the one presented here, and this is now mentioned in the revised MS - response point 7.

10) An interesting point of novelty within the current MS is the three-pronged (rather than two-pronged) nature of interaction between Rad52 and Rad51. In addition, by structural analysis/modelling, the authors succeed in providing an improved view of Rad52-Rad51 interactions. The work is solid, interesting, and should be published; however, it does not provide the kind of conceptual advance over what we know about Rad52/BRCA2-Rad51/RAD51 mediator function that would make the MS a strong candidate for publication in Nature Communications.

As already mentioned, our study not only provides robust structural data, as acknowledged by Reviewer #2, but also includes phenotypic analyses of three Rad52 mutants that subtly affect its mediator activity. Taken together, these findings support a new conceptual framework for Rad52 function: the idea that Rad52 acts as an assembly chaperone for Rad51, both promoting filament formation and protecting it from Srs2-mediated disassembly. This dual role—mechanistically defined and experimentally supported—represents a meaningful conceptual advance beyond current models of mediator function for Rad52 and BRCA2.

Reviewer #3

(Remarks to the Author)

The comments and concerns I raised in the previous round of peer review have been addressed to my satisfaction. I would recommend this paper be accepted for publication.

Rad52 Acts as an Assembly Chaperone to Form and Stabilize Rad51 Filaments Through a Large C-Terminus 85-Residue Segment

Corresponding Author: Dr Eric Coïc

Version 0:

Reviewer comments:

Reviewer #1

(Remarks to the Author)

In Ma et al. The authors deliver an excellent and timely paper discussing the ability of Rad52 to act as a molecular chaperone to Rad51. The experiments are technically well done, and the findings support the primary conclusions of the paper. The major conclusions are that a disordered surface of Rad52 interacts with the oligomerization of Rad51 to prevent self-association and promote loading on ssDNA. The authors also find that this region protects Rad51 filament disassembly by the motor protein Srs2. Although, the mechanism of this is less clear (See below). This excellent study represents a significant conceptual advance in our understanding of Rad51 filament formation. The one limitation of the study is a more defined relationship between the BRCA2 activity from higher eukaryotes and the activity of Rad52 in yeast if there were some relationship between the sequences identified that went beyond amino acid alignment, that would be cool. However, it is not necessary and in my opinion this manuscript is suitable for publication with some revision of the text. I have listed some major and minor comments below.

Major:

Rad52 and Brca2 are functional homologs for Rad51 loading only. BRCA2 lacks and annealase domain which is vital for function. This distinction is important.

This is an equilibrium problem. The evidence does not support direct protection of Rad51 by Rad52 on ssDNA. All it shows is that an equilibrium is shifted to maintain filament formation. I understand the EM images from (Ma et al. 2018) suggest direct protection. However, without a real-time observation of Rad51 filament protection, I don't think the conclusion of direct protection can be made. Similar arguments were made for Rad55/57 and Srs2 (Liu et al. 2011), and this was later found to be an EM artifact when Rad55/57 loading of Rad51 was directly visualized in real time (Roy et al. 2021). I think maybe tempering the language is appropriate here.

How well conserved are these regions with BRCA2? There is functional redundancy between the proteins, but I am wondering how that manifests on the protein level. Are there any other properties of these regions, outside of sequence, that might drive the interaction in both context? For example, are there flexibility properties that are similar but have different amino acids.

Minor:

Previously, we showed that deleting the Srs2 translocase, which efficiently displaces Rad51 from ssDNA,

Supplementary Figure 5: Please add a citation or permission for the data from a previous publication.

(Remarks to the Author)

Yeast Rad52 and its functional homolog in human, the tumour suppressor BRCA2, promote the actions of the strand exchange protein Rad51/RAD51 to mediate homologous recombination reactions. This interplay is essential for chromosome stability, remains incompletely understood, and studies in this area are topical and of great interest to the field. The ‘mediator’ role of Rad52 and BRCA2 comprises (1) the recruitment of the Rad51/RAD51 recombinase, and (2) enabling productive Rad51/RAD51 nucleoprotein filament formation on single-stranded DNA. Interactions between multiple BRCA2 ‘BRC repeats’ and RAD51 have been characterized in detail; interestingly BRC motifs dissociate RAD51 oligomers in solution by mimicking the RAD51 protomer–protomer interface, thereby competing with RAD51 oligomerization (ref. 14 in the current MS), while BRCA2 and Rad52 also stabilize RAD51/Rad51 oligomers on DNA (nucleoprotein filaments). Interactions like those between the BRCs and RAD51 occur between motifs in the C-terminal domain of Rad52 and Rad51 in yeast (e.g., ref. 53). A recent cryo-EM study has shown that Rad52 functions as a homodecamer, where the structured N-terminal domain of each subunit forms one unit of a ring structure. The respective C-terminal half of each subunit, which contain the binding sites for Rad51 as well as for single-stranded DNA binding protein RPA, remained disordered and structurally intractable (ref. 25).

The current study “Rad52 Acts as an Assembly Chaperone to Form and Stabilize Rad51 Filaments Through a Large C-Terminus 85-Residue Segment” by Emilie Ma and co-workers addresses the details of the Rad52 C-terminal interactions with Rad51 in *Saccharomyces cerevisiae*, extending on a previous report (ref. 39). The authors demonstrate a three-pronged (as opposed to a previously suggested two-pronged) interaction between the Rad52 C-terminus and Rad51 and provide a comprehensive mutational analysis to deduce the relative importance of each touchpoint for Rad52 mediator function. There are a few specific points which could be improved, for example including western blots to accompany the quantifications of the Rad52-Rad51 co-IP data (e.g., in Supplementary Figure S3G). It is also not clear how the authors rationalize that rad51-T161R mutant shows complete loss of Rad52 binding yet only a mild MMS sensitivity phenotype (Fig. 7), whilst structure-guided point mutations in Rad52 that lose interaction with Rad51 resulted in MMS hypersensitivity (see conclusion on page 9, “Interestingly, mutants that exhibited a complete loss of interaction with Rad51, as assessed by co-immunoprecipitation (coIP), were sensitive to low doses of MMS (0.01%), highlighting the functional importance of this interaction.”). Generally speaking, the data is of high quality, the protein modelling is appropriately presented and validated, and the conclusions are sound.

Having said that, there is a major concern that the structural and functional results in this study largely confirm existing data and the conceptual advance appears to be limited.

In their preceding 2018 study (ref. 39), the authors concentrated on a region within Rad52 spanning amino acid residues 316 to 379, which contains two previously established Rad52-binding motifs (FVTA) at position 316-319 and (YEKF) at position 376-379 (defined by the Sung and Shibata/Kurumizaka groups in refs. 51 and 53). In the present MS, it is roughly the region demarcated by these known Rad52 motifs that the authors re-label as an 85 amino-acid Rad51 interaction domain (spanning residues 310-395). This domain is found to contain a third motif (residues 337-367) supporting Rad51 binding. An extensive mutational analysis paired with DNA damage-sensitivity spot assays, co-immunoprecipitation of Rad52 and Rad51, and recombination-dependent DNA repair genetic assays is largely consistent with the notion that, as predicted by AlphaFold, there are three motifs (the two previously defined FVTA and YEKF motifs, as well as a related motif starting at F337 of Rad52) mediating three-pronged Rad51 monomer interactions and supporting Rad51 recruitment and nucleoprotein filament formation on single-stranded DNA.

While the key literature is cited, the findings within this MS are not well integrated with the existing literature. For the most part, the results presented are congruent with the current consensus on Rad52’s mediator role. For example, section 1 of Results “A large segment of 85 residues in the center of the fully disordered Rad52 C-terminal domain interacts with Rad51” and Fig 1 overlap largely with findings in refs. 51 and 53. However, refs. 51 and 53 are not cited or discussed in this section, which makes it difficult for the reader to place the MS in its proper context and identify where new insight arises. A similar criticism applies to the subsequent section and Fig. 2, which also appears largely redundant with the literature (Rad52 binding to Rad51 promoting Rad51 oligomer dissociation, e.g. ref. 53). With respect to the main conclusion and working model in Fig. 8, similar models have been published previously for BRCA2 as well as Rad52, and while some of these previous publications are cited, this is not clearly discussed in the Discussion section to identify new advances specific to the current MS. A discussion of the model for Rad52-Rad51 interactions/Rad51-chaperoning presented recently by Deveryshetty and co-workers (ref. 25) is missing. The Deveryshetty model (Fig. 8 in ref. 25) is very similar to the one presented herein (Fig. 8 in the current MS). Moreover, the authors do not relate the ideas illustrated in their model (Fig. 8) to the suggestions of Deveryshetty and co-workers who, besides C-terminal Rad52-Rad51, also show N-terminal Rad52-Rad51 interactions that are important for Rad52’s mediator function.

Kagawa (2013) and co-workers (ref. 53) have previously noted the striking similarity between the Rad51-binding motifs in Rad52 and the RAD51-binding BRC motifs in BRCA2, and they have published a homology model for the role of the Rad52 and BRCA2 motifs in sequestering monomeric Rad51/RAD51 to mediate proper loading onto single-stranded DNA/nucleoprotein-filament nucleation and growth (see Fig. 7 in ref. 53, similar to the iteration in Supplementary Fig. 4 in the current MS). Based on their structural data, Deveryshetty and co-workers (ref. 25) envisage a Rad51 chaperone model, where the C-terminal tails of Rad52 extending from the Rad52 homodecameric ring (formed by the Rad52 N-terminal domains) sequester Rad51 molecules and deliver them for assembly onto DNA (ref. 25).

An interesting point of novelty within the current MS is the three-pronged (rather than two-pronged) nature of interaction

between Rad52 and Rad51. In addition, by structural analysis/modelling, the authors succeed in providing an improved view of Rad52-Rad51 interactions. The work is solid, interesting, and should be published; however, it does not provide the kind of conceptual advance over what we know about Rad52/BRCA2-Rad51/RAD51 mediator function that would make the MS a strong candidate for publication in Nature Communications.

Reviewer #3

(Remarks to the Author)

This paper studies the structural basis for the interactions between Rad51 and Rad52, and the role of these interactions in RAD51 filament formation and filament stability. The authors have identified the region spanning residues 310-395 in the C-terminal of Rad52 as the Rad51-interacting region using NMR. They have then generated a model of the Rad51-Rad52 interaction using the NMR data and combining it with SAXS data and AlphaFold modelling. Multiple sets of experiments, both in vitro and in vivo, have then been carried out to verify this model using site directed mutants. The authors conclude that the interaction consists of three motifs, two of which bind to the Rad51 multimerization surface to prevent oligomerization and facilitate loading of monomers. The third motif is seen to stabilise Rad51 filaments by preventing disassembly by Srs2. Based on their results, they then propose a mechanistic model for the loading of Rad51 onto ssDNA by Rad52.

The study is novel and rigorous, and the molecular mechanisms involving Rad51 are important for the fields of DNA replication and repair. My concerns are listed below, and there are some minor language corrections I have annotated in the file. I recommend that this paper be accepted once these concerns are satisfactorily resolved.

Concerns:

- 1) The results section titled 'Rad52 (310-394) folds upon Rad51 binding burying a large conserved surface' describes the interaction interface of Rad51-Rad52, but this description is poorly quantified in the text and not clearly visualised in Fig. 3. An analysis of the surface to include the residue-level interactions such as H-bonding, pi-stacking should be added, in addition to the data on surface area and buried residues. The figures show Rad51 as a surface, figures with the individual interacting residues as sticks with corresponding bonds would help with visualisation. Fig. 3A has some green dotted lines in it but no explanation is given in the caption.
- 2) A similar issue is present in the section titled 'Rad52 (310-394) competes with Rad51 oligomerization'. A further description of the binding pockets of the anchors and their interactions should be added, and the Fig. 4B panels are hard to visualise due to the surface map over the ribbon diagram.
- 3) Fig. 4C shows graphs with intensity comparisons of the spectra of the Rad52 mutants bound to Rad51 vs the WT. One concern while designing mutations in proteins is if the resulting mutants are folded and stable. In this case adding the HSQC spectra of the mutants overlayed with the WT in the supplementary material will be helpful.
- 4) In supplementary Fig. 3F it is extremely hard to distinguish between the 5 models, a colour scheme with better contrast between the models would be helpful.

Version 1:

Reviewer comments:

Reviewer #1

(Remarks to the Author)

The authors have addressed all of my concerns. Congratulations on this cool study.

Reviewer #2

(Remarks to the Author)

The revised version of the MS addresses most points raised. Further textual clarifications are strongly recommended as detailed below.

Point by point answer to the reviewers' comments:

Reviewer #2 (Remarks to the Author)

Yeast Rad52 and its functional homolog in human, the tumour suppressor BRCA2, promote the actions of the strand exchange protein Rad51/RAD51 to mediate homologous recombination reactions. This interplay is essential for chromosome stability, remains incompletely understood, and studies in this area are topical and of great interest to the field. The 'mediator' role of Rad52 and BRCA2 comprises (1) the recruitment of the Rad51/RAD51 recombinase, and (2) enabling productive Rad51/RAD51 nucleoprotein filament formation on single-stranded DNA. Interactions between multiple BRCA2 'BRC repeats' and RAD51 have been characterized in detail; interestingly BRC motifs dissociate RAD51 oligomers in solution by mimicking the RAD51 protomer-protomer interface, thereby competing with RAD51 oligomerization (ref. 14 in the current MS), while BRCA2 and Rad52 also stabilize RAD51/Rad51 oligomers on DNA (nucleoproteinfilaments). Interactions like those between the BRCs and RAD51 occur between motifs in the C-terminal domain of Rad52 and Rad51

in yeast (e.g., ref. 53). A recent cryo-EM study has shown that Rad52 functions as a homodecamer, where the structured N-terminal domain of each subunit forms one unit of a ring structure. The respective C-terminal half of each subunit, which contain the binding sites for Rad51 as well as for single-stranded DNA binding protein RPA, remained disordered and structurally intractable (ref. 25).

The current study “Rad52 Acts as an Assembly Chaperone to Form and Stabilize Rad51 Filaments Through a Large C-Terminus 85-Residue Segment” by Emilie Ma and co-workers addresses the details of the Rad52 C-terminal interactions with Rad51 in *Saccharomyces cerevisiae*, extending on a previous report (ref. 39). The authors demonstrate a three-pronged (as opposed to a previously suggested two-pronged) interaction between the Rad52 C-terminus and Rad51 and provide a comprehensive mutational analysis to deduce the relative importance of each touchpoint for Rad52 mediator function.

We thank the reviewer for summarizing key background information on the topic and for highlighting previous publications, which we also cite in our manuscript. We would like to emphasize, however, that the present study goes significantly beyond our previous report (ref. 39). In that earlier work, we showed that Rad52–Rad51 interaction is not strictly required for Rad51 filament formation, we also observed that Rad52 mutants disrupting this interaction suppress filament toxicity in *srs2Δ* cells and produce filaments that are more sensitive to Srs2-mediated disassembly. These findings led us to propose that Rad52 also stabilizes Rad51 filaments, though the mechanism was unknown.

Here, we provide the first detailed structural description of the Rad52–Rad51 interface, revealing an extended 85-residue Rad52 segment using NMR, AlphaFold, and SAXS. We show that Rad52 binds a single Rad51 monomer in competition with Rad51–Rad51 oligomerization, supporting a model in which Rad52 acts as a chaperone to isolate monomers and load them onto ssDNA.

As the reviewer notes, we tested this model through a comprehensive mutational analysis. Importantly, we extended this approach to assess the role of each contact site not only in mediator activity, but also in Rad51 filament formation by live microscopy and competition with Srs2—an unexplored aspect until now.

1) There are a few specific points which could be improved, for example including western blots to accompany the quantifications of the Rad52–Rad51 co-IP data (e.g., in Supplementary Figure S3G).

We added the western blots in addition to the quantification in Supplementary Figure S3G and 3H of the revised manuscript (cited page 13 of the main text).

Referee response: The WBs are a valuable addition, although one might argue that showing immunodetection for IPed Rad51 rather than WCE Rad51 would be the appropriate control here.

2) It is also not clear how the authors rationalize that *rad51-T161R* mutant shows complete loss of Rad52 binding yet only a mild MMS sensitivity phenotype (Fig. 7), whilst structure-guided point mutations in Rad52 that lose interaction with Rad51 resulted in MMS hypersensitivity (see conclusion on page 9, “Interestingly, mutants that exhibited a complete loss of interaction with Rad51, as assessed by co-immunoprecipitation (coIP), were sensitive to low doses of MMS (0.01%), highlighting the functional importance of this interaction.”).

The sentence on page 9 cited by the reviewer refers to a general correlation between loss of Rad52–Rad51 interaction and MMS sensitivity, without yet detailing the variability among mutants. To clarify this point, we have specified page 9 in the revised manuscript that mutants losing Rad52–Rad51 interaction show MMS sensitivity to varying degrees.

As noted in the original text (page 13), “*rad52-F316A* and *F337A* (anchors 1 and 2) exhibit higher sensitivity to MMS (Figure 3D) compared to *rad52-Y376A* (anchor 3).” A similar gradient between mutants was also observed after γ -irradiation (page 13 of the original text). On page 18 we compared *rad51-T161R* mutant with *rad52-Y376A*, stating that they display similar MMS sensitivity: “...comparable to that of *rad52-Y376A* cells (Figure 7B).” To avoid any confusion, we have now explicitly added page 18 a reference to Figure 3D in the revised manuscript to clarify this comparison.

We would like to emphasize that while all three anchor mutations disrupt Rad52–Rad51 interaction (as shown by NMR and co-IP), their *in vivo* effects are more nuanced. As emphasized in the discussion, we stress the importance of combining *in vitro* and *in vivo* approaches to detect subtle phenotypes that may be influenced by additional cellular factors. The range of sensitivities observed supports the distinct functional contributions of each anchor to Rad51 filament dynamics.

Referee response: OK.

3) Generally speaking, the data is of high quality, the protein modelling is appropriately presented and validated, and the conclusions are sound. Having said that, there is a major concern that the structural and functional results in this study largely confirm existing data and the conceptual advance appears to be limited.

In their preceding 2018 study (ref. 39), the authors concentrated on a region within Rad52 spanning amino acid residues 316 to 379, which contains two previously established Rad52-binding motifs (FVTA) at position 316-319 and (YEKF) at position 376-379 (defined by the Sung and Shibata/Kurumizaka groups in refs. 51 and 53).

In the present MS, it is roughly the region demarcated by these known Rad52 motifs that the authors re-label as an 85 amino-acid Rad51 interaction domain (spanning residues 310-395). This domain is found to contain a third motif (residues 337-367) supporting Rad51 binding. An extensive mutational analysis paired with DNA damage-sensitivity spot assays, co-immunoprecipitation of Rad52 and Rad51, and recombination-dependent DNA repair genetic assays is largely consistent with the notion that, as predicted by AlphaFold, there are three motifs (the two previously defined FVTA and YEKF motifs, as well as a related motif starting at F337 of Rad52) mediating three-pronged Rad51 monomer interactions and supporting Rad51 recruitment and nucleoprotein filament formation on single-stranded DNA.

We thank the reviewer for acknowledging the quality of our data.

Contrary to what is stated by the reviewer, our 2018 study (ref. 39) focused solely on a narrow segment of Rad52 (residues 371–387), corresponding to the YEKF motif. In contrast, the present work defines and characterizes, for the first time, the entire Rad51-binding domain of Rad52 (residues 310–395), using a combination of NMR, AlphaFold modelling, SAXS, and extensive *in vivo* functional assays.

The two previously known motifs (FVTA and YEKF) were primarily characterized *in vitro* (ref. 51 and 53), with only limited *in*

vivo analysis that did not account for Srs2-dependent effects. In this study, we identify and validate a third functional Rad51-binding motif, centered on F337, and demonstrate that Rad52 uses three distinct “anchors” to engage Rad51 monomers. Notably, we show that anchors 1 and 2 compete with the Rad51–Rad51 interface, thereby directly influencing filament formation, while anchor 3 could play a specific role in protecting filaments from Srs2 disassembly, with less impact on filament assembly itself.

These findings go well beyond confirming previous data. They define a cohesive structural and functional framework for Rad52’s mediator activity, clarify the differential contributions of each interaction site to filament dynamics, and establish a mechanistic link between Rad52 architecture and its dual role in Rad51 loading and filament stabilization.

4) While the key literature is cited, the findings within this MS are not well integrated with the existing literature. For the most part, the results presented are congruent with the current consensus on Rad52’s mediator role. For example, section 1 of Results “A large segment of 85 residues in the center of the fully disordered Rad52 C-terminal domain interacts with Rad51” and Fig 1 overlap largely with findings in refs. 51 and 53. However, refs. 51 and 53 are not cited or discussed in this section, which makes it difficult for the reader to place the MS in its proper context and identify where new insight arises.

Refs. 51 (Krejci et al. 2002) and 53 (Kagawa et al. 2014) are cited in the Introduction, prior to the presentation of our results. To help readers better contextualize our findings in light of these previous studies, we have now added explicit citations to refs. 51 and 53 in line 3 of the first paragraph of the Results section (page 5).

However, we would like to emphasize that these studies did not provide structural data to analyze the Rad52–Rad51 binding mode, as we do in Figures 1 to 4. Furthermore, both refs. 51 and 53 did not consider the contribution of Srs2 activity and incorrectly concluded that Rad52–Rad51 interaction is absolutely required for Rad51 filament formation. In contrast, our results demonstrate that mutants affecting this interaction can still form Rad51 filaments, but these filaments are highly sensitive to Srs2-mediated disassembly. Again, the comprehensive approach presented in this manuscript—including structural characterization, in vitro binding assays, and in vivo phenotypic analysis—provides a more integrated and robust framework for understanding Rad52’s mediator activity and its role in regulating Rad51 filament dynamics.

Referee response: On re-reading the revised version of the MS, I still feel that previous work (ref. 51, 53) is not appropriately credited with defining the FVTA and YEKF Rad51-binding sites which are confirmed herein. The way the FVTA motif is introduced on p.4 is rather cryptic (“The Rad51 polymerization motif FVTA is conserved in Rad52 C-terminus and essential to disrupt Rad51 oligomers.”) and the sentence makes no reference to the literature or the fact that the FVTA site has been defined as a conserved Rad51-binding domain with a notable similarity to BRC domains (ref. 53). Similarly, the Sung lab has shown by sequence alignment and mutational analysis that a second Rad51-binding site exists within Rad52 (YEKF) and is conserved (ref. 51). This should be made explicitly clear in the Introduction, which will help the reader appreciate where the data presented in the current MS confirms previous findings and conclusion, and which additional insight is won herein.

Adding refs. 51 and 53 in the first para of Results (p.5; “Although the C-terminal region of Rad52 (...) mediates essential interactions with the recombinase Rad51 [refs. 51,53] (...)”) is an important improvement, but I fear the significance of this work will be lost on the reader. I strongly recommend the authors make it clear that these previous papers have defined clearly demarcated motifs that form the downstream and upstream boundaries of the 85-residue region defined in the current MS and that these motifs are equivalent to what is described here as ‘anchor 1’ and ‘anchor 3’. This will provide much better context and, I feel, takes nothing away from the discovery of ‘anchor 3’ or the structural and functional implications described in the current MS. The potential impression that specific anchor points (1 and 3) have not been previously defined at amino acid level should be avoided, in particular since historic Rad52 residue numbering in refs. 51 and 53 could make it difficult for some readers to spot the congruence between the earlier work and the current paper. Appropriate textual changes are required before publication.

5) A similar criticism applies to the subsequent section and Fig. 2, which also appears largely redundant with the literature (Rad52 binding to Rad51 promoting Rad51 oligomer dissociation, e.g. ref. 53).

While Rad52’s role in promoting Rad51 oligomer dissociation has been previously shown (e.g., ref. 53), the data presented in Figure 2 provide additional structural insight through SAXS analysis. This approach allows us to determine the stoichiometry of the complex and to compare it across various mutants under identical experimental conditions. We have now added page 7, a reference to the relevant literature in this section of the manuscript, as follows: “These results are fully consistent with previous analyses of this complex by SEC (ref. 53).”

Referee response: OK.

6) With respect to the main conclusion and working model in Fig. 8, similar models have been published previously for BRCA2 as well as Rad52, and while some of these previous publications are cited, this is not clearly discussed in the Discussion section to identify new advances specific to the current MS

Regarding the similarities and differences between BRCA2 and Rad52, the revised manuscript now includes a more complete, dedicated paragraph at the end of the Discussion section, in response to Reviewer #1’s comment. This paragraph discusses their respective roles in Rad51 filament stability and highlights that anchor 3, which contributes specifically to filament protection, is absent from BRCA2. This underscores the unique ability of Rad52 to bind Rad51 through an extended interaction segment comprising three distinct regions, each exerting a subtle but specific impact on Rad52’s mediator activity.

Referee response: OK.

7) A discussion of the model for Rad52-Rad51 interactions/Rad51-chaperoning presented recently by Deveryshetty and co-workers (ref. 25) is missing. The Deveryshetty model (Fig. 8 in ref. 25) is very similar to the one presented herein (Fig. 8 in the current MS). Moreover, the authors do not relate the ideas illustrated in their model (Fig. 8) to the suggestions of

Deveryshetty and co-workers who, besides C-terminal Rad52-Rad51, also show N-terminal Rad52-Rad51 interactions that are important for Rad52's mediator function.

We agree that a comparative discussion of our model with the one proposed by Deveryshetty et al. (ref. 25) is a valuable addition to the manuscript. We have therefore added, a sentence in the Discussion page 21 section that refers directly to this study: "This oligomerization ability is conferred by its N-terminal annealase domain, whose structure was recently solved (ref. 25) (Supplementary Figure 1A). In that study, the authors proposed that the ssDNA-binding property of this domain, together with the association of Rad51 with the Rad52 C-terminal domain, could facilitate the transfer of Rad51 onto single-stranded DNA.". Ref. 25 is also now cited at the end of the Discussion about the multisite organization of Rad52 and BRCA2 mediators. Also we gave more details about the model in the legend of Figure 8.

Importantly, our model differs substantially from that of Deveryshetty et al., both in focus and in the nature of the supporting data. Their study primarily addresses the structural characterization of the N-terminal annealase domain and includes the C-terminal region in their model, but without experimental validation. In a complementary manner, our study focuses on the C-terminal domain of Rad52, and we integrate relevant insights from ref. 25 into our model. However, we do not attempt to depict interactions between the annealase domain and DNA, as this falls outside the scope of our experimental data.

Referee response: OK.

8) Kagawa (2013) and co-workers (ref. 53) have previously noted the striking similarity between the Rad51-binding motifs in Rad52 and the RAD51-binding BRC motifs in BRCA2, and they have published a homology model for the role of the Rad52 and BRCA2 motifs in sequestering monomeric Rad51/RAD51 to mediate proper loading onto single-stranded DNA/nucleoprotein-filament nucleation and growth (see Fig. 7 in ref. 53, similar to the iteration in Supplementary Fig. 4 in the current MS).

Indeed, the very interesting article published by Kagawa and co-workers (ref. 53) identified a clear similarity between the FxxA motifs of Rad52 and BRCA2. Figure 7 of that study presents the structure of Rad51 in complex with BRC4 by Pellegrini et al (ref 14 in our manuscript), and a sequence alignment of Rad52 homologues and, based on these information, this paper proposes that the Y376EKF motif in Rad52 could bind Rad51 similarly to F1524 in the BRC4 motif of BRCA2. However, this homology-based model is incorrect. Our structural analysis demonstrates that Y376 binds to a completely different site on Rad51 compared to F1524. To clarify this point, we have added Supplementary Figure 8, which directly compares the binding modes. Our study thus resolves a key inaccurate hypothesis from ref. 53, which arose due to the absence of reliable structural data in that earlier work.

Referee response: The conceptual point made by Kagawa et al. should be acknowledged in this context, with citation of ref. 53, regardless of whether their working hypothesis in the 2013 paper faithfully reflects the detailed protein interaction or not. I can't see where this is acknowledged, but if not done already, the insightful prediction by Kagawa et al. should be mentioned.

9) Based on their structural data, Deveryshetty and co-workers (ref. 25) envisage a Rad51 chaperone model, where the C-terminal tails of Rad52 extending from the Rad52 homodecameric ring (formed by the Rad52 N-terminal domains) sequester Rad51 molecules and deliver them for assembly onto DNA (ref. 25).

The model proposed by Deveryshetty et al. (ref. 25) suggests that each C-terminal tail of Rad52 binds two Rad51 molecules—an assumption not supported by experimental data. Furthermore, their study does not propose a chaperone model per se, but rather a scaffolding model in which each C-terminal region recruits multiple Rad51 molecules to generate a sufficient local concentration for filament nucleation. Notably, the concept of a Rad51 chaperone is never explicitly mentioned in the text of their paper.

Referee response: Semantics aside, the Deveryshetty model is conceptually very similar to the one presented here, and this is now mentioned in the revised MS - response point 7.

10) An interesting point of novelty within the current MS is the three-pronged (rather than two-pronged) nature of interaction between Rad52 and Rad51. In addition, by structural analysis/modelling, the authors succeed in providing an improved view of Rad52-Rad51 interactions. The work is solid, interesting, and should be published; however, it does not provide the kind of conceptual advance over what we know about Rad52/BRCA2-Rad51/RAD51 mediator function that would make the MS a strong candidate for publication in Nature Communications.

As already mentioned, our study not only provides robust structural data, as acknowledged by Reviewer #2, but also includes phenotypic analyses of three Rad52 mutants that subtly affect its mediator activity. Taken together, these findings support a new conceptual framework for Rad52 function: the idea that Rad52 acts as an assembly chaperone for Rad51, both promoting filament formation and protecting it from Srs2-mediated disassembly. This dual role—mechanistically defined and experimentally supported—represents a meaningful conceptual advance beyond current models of mediator function for Rad52 and BRCA2.

Reviewer #3

(Remarks to the Author)

The comments and concerns I raised in the previous round of peer review have been addressed to my satisfaction. I would recommend this paper be accepted for publication.

Point by point answer to the reviewers' comments:

Reviewer #1 (Remarks to the Author)

In Ma et al. The authors deliver an excellent and timely paper discussing the ability of Rad52 to act as a molecular chaperone to Rad51. The experiments are technically well done, and the findings support the primary conclusions of the paper. The major conclusions are that a disordered surface of Rad52 interacts with the oligomerization of Rad51 to prevent self-association and promote loading on ssDNA. The authors also find that this region protects Rad51 filament disassembly by the motor protein Srs2. Although, the mechanism of this is less clear (See below). This excellent study represents a significant conceptual advance in our understanding of Rad51 filament formation. The one limitation of the study is a more defined relationship between the BRCA2 activity from higher eukaryotes and the activity of Rad52 in yeast if there were some relationship between the sequences identified that went beyond amino acid alignment, that would be cool. However, it is not necessary and in my opinion this manuscript is suitable for publication with some revision of the text. I have listed some major and minor comments below.

We appreciate the reviewer's positive comments on our manuscript, particularly regarding the timeliness of the study. We have developed a comparative analysis of Rad52 and BRCA2 binding modes with Rad51, which we detail below.

Major:

- 1) Rad52 and Brca2 are functional homologs for Rad51 loading only. BRCA2 lacks and annealase domain which is vital for function. This distinction is important.

We fully agree with the reviewer that BRCA2 lacks an annealing domain and that it is important to make this precision. We have changed the second paragraph on page 3 to emphasize this difference.

- 2) This is an equilibrium problem. The evidence does not support direct protection of Rad51 by Rad52 on ssDNA. All it shows is that an equilibrium is shifted to maintain filament formation. I understand the EM images from (Ma et al. 2018) suggest direct protection. However, without a real-time observation of Rad51 filament protection, I don't think the conclusion of direct protection can be made. Similar arguments were made for Rad55/57 and Srs2 (Liu et al. 2011), and this was later found to be an EM artifact when Rad55/57 loading of Rad51 was directly visualized in real time (Roy et al. 2021). I think maybe tempering the language is appropriate here.

Our working model is based on the hypothesis that the Rad51-Rad52 interaction not only promotes Rad51 filament formation but also directly influences its stability. This model is supported by converging evidence cited by the reviewer (Ma et al., eLife 2018). This work provides further support for this model, as the binding of anchor 3 does not compete with the Rad51 oligomerization interface on DNA and is therefore compatible with binding to Rad51 filaments. This configuration provides a structural explanation for the potential protection of the Rad51 filament by Rad52 against Srs2 activity. However, we acknowledge that we do not provide definitive biochemical proof of such a direct protective action. Our observations could indeed result from a shift in reaction equilibria favoring filament formation over dissociation. Accordingly, we have tempered our language in several sections of the text as detailed below (Abstract, Introduction pp. 3-5, Results pp. 13, 14 and 18, Discussion pp. 20 and 24).

Abstract : “These findings highlight how Rad52 functions as an assembly chaperone, preventing Rad51 oligomerization, promoting nucleation of Rad51 nucleofilaments on ssDNA, and protecting Rad51 filaments from destabilization by Srs2.” was replaced by “These findings highlight how Rad52 functions as an assembly chaperone by preventing Rad51 oligomerization, promoting the nucleation of Rad51 nucleofilaments on ssDNA, and

counteracting the effect of Srs2 on the destabilization of Rad51 filament destabilization.”

Introduction page 3: “Until recently, Rad52 was mainly seen as a mediator promoting the replacement of RPA by Rad51. However, there is now increasing evidence for other roles in Rad51 filament formation and stability. In particular, we have shown that Rad52 is essential for protecting Rad51 filaments from dissociation by the Srs2 DNA translocase³⁹” was replaced by “Rad52 was primarily viewed as a mediator promoting the replacement of RPA by Rad51. There is now increasing evidence for additional roles of Rad52 in Rad51 filament formation and stability. In particular, we have gathered evidence suggesting that Rad52 might also protect Rad51 filaments from dissociation by the Srs2 DNA translocase⁴¹”.

Introduction page 4: “In addition, the interaction between Rad51 and the C-terminal domain of Rad52 protects the filaments from dissociation by the Srs2 helicase^{39,58}. However, in Srs2-deficient cells, this interaction makes Rad51 filaments toxic, probably because it confers excessive stability, resulting in the accumulation of aberrant Rad51 filaments that obstruct DNA metabolism⁸” was replaced by “We found that Rad52 mutants affecting Rad51 interaction are still proficient in Rad51 filament formation, but the filaments formed are more sensitive to Srs2 helicase activity^{41,58}. Furthermore, in Srs2-deficient cells with strong native Rad52-Rad51 interactions, we observed the accumulation of aberrant Rad51 filaments leading to severe toxicity, probably due to obstructions in DNA metabolisms⁸. Therefore, our current model states that Rad51 filament homeostasis is finely regulated by the interplay between Rad52 and Srs2.”

Introduction page 4-5: “We propose that this segment of Rad52 functions as an assembly chaperone, preventing Rad51 oligomerization and promoting the formation of a functional nucleofilament. Furthermore, the entire segment plays a role in protecting Rad51 filaments from destabilization by Srs2.” was replaced by “We propose that this segment of Rad52 functions as an assembly chaperone, preventing Rad51 oligomerization, promoting the formation of a functional nucleofilament, and counteracting the destabilization action of Srs2.”

Results Page 13: “is essential for protecting Rad51 filaments against Srs2.” was replaced by “is essential for the destabilization of Rad51 filaments by Srs2..”

Results Page 14: “but is deficient in protecting filaments from Srs2.” was replaced by “but is poorly able to compete with Srs2.”

Discussion Page 20: “We show that Anchor 3 in this motif is important for stabilizing Rad51 filaments and protecting them from Srs2,” was replaced by “We show that Anchor 3 in this motif is important for Rad51 filaments to better resist Srs2,”

Discussion Page 20: “This indicates that these residues contribute to Rad51 filament formation to varying degrees and appear to be important for Rad51 filament protection.” was replaced by “suggesting that each of them contributes to Rad51 filament formation to varying degrees and plays a role in filament homeostasis.”

Discussion Page 22: “Rad52 also binds Rad51 to stabilize Rad51 by association with filaments” was replaced by “Rad52 also binds Rad51 to efficiently compete with Srs2”

Discussion Page 22: “However, the very similar phenotypes of the *rad51-T161R* mutation, where Rad51-T161 faces Rad52-Y376 in the binding site, strongly support an involvement of the third motif in protecting the Rad51 filament from Srs2 rather than in Rad51 filament formation.” was replaced by “However, the very similar phenotypes of the *rad51-T161R*

and *Rad52-Y376* mutations, together with the fact that *Rad51-T161* and *Rad52-Y376* face each other at the interface, strongly support the idea that anchor 3 is particularly important to compete with *Srs2* activity, while it is less critical for *Rad51* filament formation.”

- 3) How well conserved are these regions with *BRCA2*? There is functional redundancy between the proteins, but I am wondering how that manifests on the protein level. Are there any other properties of these regions, outside of sequence, that might drive the interaction in both context? For example, are there flexibility properties that are similar but have different amino acids.

We appreciate the reviewer’s suggestions and have expanded the corresponding paragraph in the Discussion to include a more complete discussion about conserved features between *BRCA2* and *Rad52*. We have added a Supplementary Figure (**Supplementary Figure 8**) showing the sequence conservation of the eight BRC repeats from human *BRCA2* in the form of WebLogos, as well as the superposition of AlphaFold-predicted structures for these eight BRC motifs bound to *Rad51*. These predicted structures are highly similar to the experimentally determined structure of mouse *BRC4* motif bound to *Rad51*. This binding mode is compared to that of *Rad52* in a view with the same orientation, clearly illustrating that the binding modes differ—except for the short *FxxA* motif illustrated in **Supplementary Figure 4A**. Additionally, we have added a paragraph discussing structural features shared between *BRCA2* and the *Rad52* C-terminal region, independently of their primary sequence (page 23-24 of the revised manuscript).

Minor:

- 1) Previously, we showed that deleting the *Srs2* translocase, which efficiently displaces *Rad51* from ssDNA,

This has been corrected

- 2) Supplementary Figure 5: Please add a citation or permission for the data from a previous publication.

A reference has been added in the legend as well as in the legend of Figure 5

Reviewer #2 (Remarks to the Author)

Yeast *Rad52* and its functional homolog in human, the tumour suppressor *BRCA2*, promote the actions of the strand exchange protein *Rad51/RAD51* to mediate homologous recombination reactions. This interplay is essential for chromosome stability, remains incompletely understood, and studies in this area are topical and of great interest to the field. The ‘mediator’ role of *Rad52* and *BRCA2* comprises (1) the recruitment of the *Rad51/RAD51* recombinase, and (2) enabling productive *Rad51/RAD51* nucleoprotein filament formation on single-stranded DNA. Interactions between multiple *BRCA2* ‘BRC repeats’ and *RAD51* have been characterized in detail; interestingly BRC motifs dissociate *RAD51* oligomers in solution by mimicking the *RAD51* protomer–protomer interface, thereby competing with *RAD51* oligomerization (ref. 14 in the current MS), while *BRCA2* and *Rad52* also stabilize *RAD51/Rad51* oligomers on DNA (nucleoproteinfilaments). Interactions like those between the BRCs and *RAD51* occur between motifs in the C-terminal domain of *Rad52* and *Rad51* in yeast (e.g., ref. 53). A recent cryo-EM study has shown that *Rad52* functions as a homodecamer, where the structured N-terminal domain of each subunit forms one unit of a ring structure. The respective C-terminal half of each subunit, which contain the binding sites for *Rad51* as

well as for single-stranded DNA binding protein RPA, remained disordered and structurally intractable (ref. 25).

The current study “Rad52 Acts as an Assembly Chaperone to Form and Stabilize Rad51 Filaments Through a Large C-Terminus 85-Residue Segment” by Emilie Ma and co-workers addresses the details of the Rad52 C-terminal interactions with Rad51 in *Saccharomyces cerevisiae*, extending on a previous report (ref. 39). The authors demonstrate a three-pronged (as opposed to a previously suggested two-pronged) interaction between the Rad52 C-terminus and Rad51 and provide a comprehensive mutational analysis to deduce the relative importance of each touchpoint for Rad52 mediator function.

We thank the reviewer for summarizing key background information on the topic and for highlighting previous publications, which we also cite in our manuscript. We would like to emphasize, however, that the present study goes significantly beyond our previous report (ref. 39). In that earlier work, we showed that Rad52–Rad51 interaction is not strictly required for Rad51 filament formation, we also observed that Rad52 mutants disrupting this interaction suppress filament toxicity in *srs2Δ* cells and produce filaments that are more sensitive to Srs2-mediated disassembly. These findings led us to propose that Rad52 also stabilizes Rad51 filaments, though the mechanism was unknown.

Here, we provide the first detailed structural description of the Rad52–Rad51 interface, revealing an extended 85-residue Rad52 segment using NMR, AlphaFold, and SAXS. We show that Rad52 binds a single Rad51 monomer in competition with Rad51–Rad51 oligomerization, supporting a model in which Rad52 acts as a chaperone to isolate monomers and load them onto ssDNA.

As the reviewer notes, we tested this model through a comprehensive mutational analysis. Importantly, we extended this approach to assess the role of each contact site not only in mediator activity, but also in Rad51 filament formation by live microscopy and competition with Srs2—an unexplored aspect until now.

- 1) There are a few specific points which could be improved, for example including western blots to accompany the quantifications of the Rad52-Rad51co-IP data (e.g., in Supplementary Figure S3G).

We added the western blots in addition to the quantification in **Supplementary Figure S3G and 3H** of the revised manuscript (cited page 13 of the main text).

- 2) It is also not clear how the authors rationalize that *rad51-T161R* mutant shows complete loss of Rad52 binding yet only a mild MMS sensitivity phenotype (Fig. 7), whilst structure-guided point mutations in Rad52 that lose interaction with Rad51 resulted in MMS hypersensitivity (see conclusion on page 9, “Interestingly, mutants that exhibited a complete loss of interaction with Rad51, as assessed by co-immunoprecipitation (coIP), were sensitive to low doses of MMS (0.01%), highlighting the functional importance of this interaction.”).

The sentence on page 9 cited by the reviewer refers to a general correlation between loss of Rad52–Rad51 interaction and MMS sensitivity, without yet detailing the variability among mutants. To clarify this point, we have specified page 9 in the revised manuscript that mutants losing Rad52–Rad51 interaction show MMS sensitivity **to varying degrees**.

As noted in the original text (page 13), “*rad52-F316A and F337A (anchors 1 and 2) exhibit higher sensitivity to MMS (Figure 3D) compared to rad52-Y376A (anchor 3).*” A similar gradient between mutants was also observed after γ -irradiation (page 13 of the original text). On page 18 we compared *rad51-T161R* mutant with *rad52-Y376A*, stating that they display similar MMS sensitivity: “...comparable to that of *rad52-Y376A* cells (**Figure 7B**).” To avoid any confusion, we have now explicitly added page 18 a reference to **Figure 3D** in the revised manuscript to clarify this comparison.

We would like to emphasize that while all three anchor mutations disrupt Rad52–Rad51 interaction (as shown by NMR and co-IP), their *in vivo* effects are more nuanced. As emphasized in the discussion,

we stress the importance of combining *in vitro* and *in vivo* approaches to detect subtle phenotypes that may be influenced by additional cellular factors. The range of sensitivities observed supports the distinct functional contributions of each anchor to Rad51 filament dynamics.

- 3) Generally speaking, the data is of high quality, the protein modelling is appropriately presented and validated, and the conclusions are sound. Having said that, there is a major concern that the structural and functional results in this study largely confirm existing data and the conceptual advance appears to be limited.

In their preceding 2018 study (ref. 39), the authors concentrated on a region within Rad52 spanning amino acid residues 316 to 379, which contains two previously established Rad52-binding motifs (FVTA) at position 316-319 and (YEKF) at position 376-379 (defined by the Sung and Shibata/Kurumizaka groups in refs. 51 and 53).

In the present MS, it is roughly the region demarcated by these known Rad52 motifs that the authors re-label as an 85 amino-acid Rad51 interaction domain (spanning residues 310-395). This domain is found to contain a third motif (residues 337-367) supporting Rad51 binding. An extensive mutational analysis paired with DNA damage-sensitivity spot assays, co-immunoprecipitation of Rad52 and Rad51, and recombination-dependent DNA repair genetic assays is largely consistent with the notion that, as predicted by AlphaFold, there are three motifs (the two previously defined FVTA and YEKF motifs, as well as a related motif starting at F337 of Rad52) mediating three-pronged Rad51 monomer interactions and supporting Rad51 recruitment and nucleoprotein filament formation on single-stranded DNA.

We thank the reviewer for acknowledging the quality of our data.

Contrary to what is stated by the reviewer, our 2018 study (ref. 39) focused solely on a narrow segment of Rad52 (residues 371–387), corresponding to the YEKF motif. In contrast, the present work defines and characterizes, for the first time, the entire Rad51-binding domain of Rad52 (residues 310–395), using a combination of NMR, AlphaFold modelling, SAXS, and extensive *in vivo* functional assays.

The two previously known motifs (FVTA and YEKF) were primarily characterized *in vitro* (ref. 51 and 53), with only limited *in vivo* analysis that did not account for Srs2-dependent effects. In this study, we identify and validate a third functional Rad51-binding motif, centered on F337, and demonstrate that Rad52 uses three distinct “anchors” to engage Rad51 monomers. Notably, we show that anchors 1 and 2 compete with the Rad51–Rad51 interface, thereby directly influencing filament formation, while anchor 3 could play a specific role in protecting filaments from Srs2 disassembly, with less impact on filament assembly itself.

These findings go well beyond confirming previous data. They define a cohesive structural and functional framework for Rad52’s mediator activity, clarify the differential contributions of each interaction site to filament dynamics, and establish a mechanistic link between Rad52 architecture and its dual role in Rad51 loading and filament stabilization.

- 4) While the key literature is cited, the findings within this MS are not well integrated with the existing literature. For the most part, the results presented are congruent with the current consensus on Rad52’s mediator role. For example, section 1 of Results “A large segment of 85 residues in the center of the fully disordered Rad52 C-terminal domain interacts with Rad51” and Fig 1 overlap largely with findings in refs. 51 and 53. However, refs. 51 and 53 are not cited or discussed in this section, which makes it difficult for the reader to place the MS in its proper context and identify where new insight arises.

Refs. 51 (Krejci et al. 2002) and 53 (Kagawa et al. 2014) are cited in the Introduction, prior to the presentation of our results. To help readers better contextualize our findings in light of these previous studies, we have now added explicit citations to refs. 51 and 53 in line 3 of the first paragraph of the Results section (page 5).

However, we would like to emphasize that these studies did not provide structural data to analyze the Rad52–Rad51 binding mode, as we do in Figures 1 to 4. Furthermore, both refs. 51 and 53 did not consider the contribution of Srs2 activity and incorrectly concluded that Rad52–Rad51 interaction is absolutely required for Rad51 filament formation. In contrast, our results demonstrate that mutants affecting this interaction can still form Rad51 filaments, but these filaments are highly sensitive to Srs2-mediated disassembly. Again, the comprehensive approach presented in this manuscript—including structural characterization, *in vitro* binding assays, and *in vivo* phenotypic analysis—provides a more integrated and robust framework for understanding Rad52’s mediator activity and its role in regulating Rad51 filament dynamics.

- 5) A similar criticism applies to the subsequent section and Fig. 2, which also appears largely redundant with the literature (Rad52 binding to Rad51 promoting Rad51 oligomer dissociation, e.g. ref. 53).

While Rad52’s role in promoting Rad51 oligomer dissociation has been previously shown (e.g., ref. 53), the data presented in Figure 2 provide additional structural insight through SAXS analysis. This approach allows us to determine the stoichiometry of the complex and to compare it across various mutants under identical experimental conditions. We have now added page 7, a reference to the relevant literature in this section of the manuscript, as follows: “These results are fully consistent with previous analyses of this complex by SEC (ref. 53).”

- 6) With respect to the main conclusion and working model in Fig. 8, similar models have been published previously for BRCA2 as well as Rad52, and while some of these previous publications are cited, this is not clearly discussed in the Discussion section to identify new advances specific to the current MS

Regarding the similarities and differences between BRCA2 and Rad52, the revised manuscript now includes a more complete, dedicated paragraph at the end of the Discussion section, in response to Reviewer #1’s comment. This paragraph discusses their respective roles in Rad51 filament stability and highlights that anchor 3, which contributes specifically to filament protection, is absent from BRCA2. This underscores the unique ability of Rad52 to bind Rad51 through an extended interaction segment comprising three distinct regions, each exerting a subtle but specific impact on Rad52’s mediator activity.

- 7) A discussion of the model for Rad52-Rad51 interactions/Rad51-chaperoning presented recently by Deveryshetty and co-workers (ref. 25) is missing. The Deveryshetty model (Fig. 8 in ref. 25) is very similar to the one presented herein (Fig. 8 in the current MS). Moreover, the authors do not relate the ideas illustrated in their model (Fig. 8) to the suggestions of Deveryshetty and co-workers who, besides C-terminal Rad52-Rad51, also show N-terminal Rad52-Rad51 interactions that are important for Rad52’s mediator function.

We agree that a comparative discussion of our model with the one proposed by Deveryshetty et al. (ref. 25) is a valuable addition to the manuscript. We have therefore added, a sentence in the Discussion page 21 section that refers directly to this study: “*This oligomerization ability is conferred by its N-terminal annealase domain, whose structure was recently solved (ref. 25) (Supplementary Figure 1A). In that study, the authors proposed that the ssDNA-binding property of this domain, together with the association of Rad51 with the Rad52 C-terminal domain, could facilitate the transfer of Rad51 onto single-stranded DNA.*”. Ref. 25 is also now cited at the end of the Discussion about the multisite organization of Rad52 and BRCA2 mediators. Also we gave more details about the model in the legend of **Figure 8**.

Importantly, our model differs substantially from that of Deveryshetty et al., both in focus and in the nature of the supporting data. Their study primarily addresses the structural characterization of the N-terminal annealase domain and includes the C-terminal region in their model, but without

experimental validation. In a complementary manner, our study focuses on the C-terminal domain of Rad52, and we integrate relevant insights from ref. 25 into our model. However, we do not attempt to depict interactions between the annealase domain and DNA, as this falls outside the scope of our experimental data.

- 8) Kagawa (2013) and co-workers (ref. 53) have previously noted the striking similarity between the Rad51-binding motifs in Rad52 and the RAD51-binding BRC motifs in BRCA2, and they have published a homology model for the role of the Rad52 and BRCA2 motifs in sequestering monomeric Rad51/RAD51 to mediate proper loading onto single-stranded DNA/nucleoprotein-filament nucleation and growth (see Fig. 7 in ref. 53, similar to the iteration in Supplementary Fig. 4 in the current MS).

Indeed, the very interesting article published by Kagawa and co-workers (ref. 53) identified a clear similarity between the FxxA motifs of Rad52 and BRCA2. Figure 7 of that study presents the structure of Rad51 in complex with BRC4 by Pellegrini et al (ref 14 in our manuscript), and a sequence alignment of Rad52 homologues and, based on these information, this paper proposes that the Y₃₇₆EKF motif in Rad52 could bind Rad51 similarly to F1524 in the BRC4 motif of BRCA2. However, this homology-based model is incorrect. Our structural analysis demonstrates that Y376 binds to a completely different site on Rad51 compared to F1524. To clarify this point, we have added **Supplementary Figure 8**, which directly compares the binding modes. Our study thus resolves a key inaccurate hypothesis from ref. 53, which arose due to the absence of reliable structural data in that earlier work.

- 9) Based on their structural data, Deveryshetty and co-workers (ref. 25) envisage a Rad51 chaperone model, where the C-terminal tails of Rad52 extending from the Rad52 homodecameric ring (formed by the Rad52 N-terminal domains) sequester Rad51 molecules and deliver them for assembly onto DNA (ref. 25).

The model proposed by Deveryshetty et al. (ref. 25) suggests that each C-terminal tail of Rad52 binds two Rad51 molecules—an assumption not supported by experimental data. Furthermore, their study does not propose a chaperone model per se, but rather a scaffolding model in which each C-terminal region recruits multiple Rad51 molecules to generate a sufficient local concentration for filament nucleation. Notably, the concept of a Rad51 chaperone is never explicitly mentioned in the text of their paper.

- 10) An interesting point of novelty within the current MS is the three-pronged (rather than two-pronged) nature of interaction between Rad52 and Rad51. In addition, by structural analysis/modelling, the authors succeed in providing an improved view of Rad52-Rad51 interactions. The work is solid, interesting, and should be published; however, it does not provide the kind of conceptual advance over what we know about Rad52/BRCA2-Rad51/RAD51 mediator function that would make the MS a strong candidate for publication in Nature Communications.

As already mentioned, our study not only provides robust structural data, as acknowledged by Reviewer #2, but also includes phenotypic analyses of three Rad52 mutants that subtly affect its mediator activity. Taken together, these findings support a new conceptual framework for Rad52 function: the idea that Rad52 acts as an assembly chaperone for Rad51, both promoting filament formation and protecting it from Srs2-mediated disassembly. This dual role—mechanistically defined and experimentally supported—represents a meaningful conceptual advance beyond current models of mediator function for Rad52 and BRCA2.

Reviewer #3 (Remarks to the Author):

This paper studies the structural basis for the interactions between Rad51 and Rad52, and the role of these interactions in RAD51 filament formation and filament stability. The authors have identified the region spanning residues 310-395 in the C-terminal of Rad52 as the Rad51-interacting region using NMR. They have then generated a model of the Rad51-Rad52 interaction using the NMR data and combining it with SAXS data and AlphaFold modelling. Multiple sets of experiments, both in vitro and in vivo, have then been carried out to verify this model using site directed mutants. The authors conclude that the interaction consists of three motifs, two of which bind to the Rad51 multimerization surface to prevent oligomerization and facilitate loading of monomers. The third motif is seen to stabilise Rad51 filaments by preventing disassembly by Srs2. Based on their results, they then propose a mechanistic model for the loading of Rad51 onto ssDNA by Rad52.

The study is novel and rigorous, and the molecular mechanisms involving Rad51 are important for the fields of DNA replication and repair. My concerns are listed below, and there are some minor language corrections I have annotated in the file. I recommend that this paper be accepted once these concerns are satisfactorily resolved.

Concerns:

- 1) The results section titled 'Rad52 (310-394) folds upon Rad51 binding burying a large conserved surface' describes the interaction interface of Rad51-Rad52, but this description is poorly quantified in the text and not clearly visualised in Fig. 3. An analysis of the surface to include the residue-level interactions such as H-bonding, pi-stacking should be added, in addition to the data on surface area and buried residues. The figures show Rad51 as a surface, figures with the individual interacting residues as sticks with corresponding bonds would help with visualisation. Fig. 3A has some green dotted lines in it but no explanation is given in the caption.

We thank the reviewer for the constructive suggestion aimed at improving the description and visualization of the Rad52-Rad51 interface. To provide a more complete figure, including the Rad51 residues interacting with Rad52, we have added four close-up views of the interface in **Supplementary Figure 3C** of the revised manuscript, where the side chains of the interacting residues are shown. The Rad51 residues in direct contact with the three anchor residues—F316, F337, and Y376—are labeled. We have also updated the legend of **Figure 3A** to specify the type of contacts represented by green dashed lines (hydrogen bonds) page 10. In addition, we have added a supplementary table (**Supplementary Table 1** in the revised manuscript) listing all the intermolecular contacts between Rad52 and Rad51 in the model, along with an analysis of the type of interaction (hydrophobic, hydrogen bond, salt bridge, or π -stacking).

- 2) A similar issue is present in the section titled 'Rad52 (310-394) competes with Rad51 oligomerization'. A further description of the binding pockets of the anchors and their interactions should be added, and the Fig. 4B panels are hard to visualise due to the surface map over the ribbon diagram.

To provide a complete description and comparison of the binding pockets between Rad51-Rad51 and Rad52-Rad51 interfaces, we have added a supplementary table (**Supplementary Table 2** in the revised manuscript) listing all the intermolecular contacts between two Rad51 molecules in the oligomer (PDB), along with an analysis of the type of interaction (hydrophobic, hydrogen bond, salt bridge, or π -stacking).

- 3) Fig. 4C shows graphs with intensity comparisons of the spectra of the Rad52 mutants bound to Rad51 vs the WT. One concern while designing mutations in proteins is if the resulting mutants are folded and stable. In this case adding the HSQC spectra of the mutants overlaid with the WT in the supplementary material will be helpful.

We have added HSQC spectra of the four mutants, recorded both alone and after the addition of Rad51, with complete resonance assignment (**Supplementary Figure 4B** in the revised manuscript). When comparing these spectra to that of the wild-type Rad52(295–394) segment alone (**Supplementary Figure 1D**), nearly all signals are superimposable, except for the mutated residues and their immediate neighbors.

4) In supplementary Fig. 3F it is extremely hard to distinguish between the 5 models, a colour scheme with better contrast between the models would be helpful.

A revised version of **Supplementary Figure 3F** is presented, showing the top five superimposed models, and without the initial AlphaFold model for improved clarity. A color gradient from blue to green is used for Rad51, and shades of pink, purple, and orange are used for Rad52.

We thank the reviewers for their time and comments.

Reviewer #1 (Remarks to the Author):

The authors have addressed all of my concerns. Congratulations on this cool study.

Authors response:

We sincerely thank the reviewer for their valuable feedback, which helped improve the quality of the manuscript.

Reviewer #2 (Remarks to the Author):

The revised version of the MS addresses most points raised. Further textual clarifications are strongly recommended as detailed below.

Point by point answer to the reviewers' comments:

Reviewer #2 (Remarks to the Author)

Yeast Rad52 and its functional homolog in human, the tumour suppressor BRCA2, promote the actions of the strand exchange protein Rad51/RAD51 to mediate homologous recombination reactions. This interplay is essential for chromosome stability, remains incompletely understood, and studies in this area are topical and of great interest to the field. The 'mediator' role of Rad52 and BRCA2 comprises (1) the recruitment of the Rad51/RAD51 recombinase, and (2) enabling productive Rad51/RAD51 nucleoprotein filament formation on single-stranded DNA. Interactions between multiple BRCA2 'BRC repeats' and RAD51 have been characterized in detail; interestingly BRC motifs dissociate RAD51 oligomers in solution by mimicking the RAD51 protomer-protomer interface, thereby competing with RAD51 oligomerization (ref. 14 in the current MS), while BRCA2 and Rad52 also stabilize RAD51/Rad51 oligomers on DNA (nucleoprotein filaments). Interactions like those between the BRCs and RAD51 occur between motifs in the C-terminal domain of Rad52 and Rad51 in yeast (e.g., ref. 53). A recent cryo-EM study has shown that Rad52 functions as a homodecamer, where the structured N-terminal domain of each subunit forms one unit of a ring structure. The respective C-terminal half of each subunit, which contain the binding sites for Rad51 as well as for single-stranded DNA binding protein RPA, remained disordered and structurally intractable (ref. 25).

The current study "Rad52 Acts as an Assembly Chaperone to Form and Stabilize Rad51 Filaments Through a Large C-Terminus 85-Residue Segment" by Emilie Ma and co-workers addresses the details of the Rad52 C-terminal interactions with Rad51 in *Saccharomyces cerevisiae*, extending on a previous report (ref. 39). The authors demonstrate a three-pronged (as opposed to a previously suggested two-pronged) interaction between the Rad52 C-terminus and Rad51 and provide a comprehensive mutational analysis to deduce the relative importance of each touchpoint for Rad52 mediator function.

We thank the reviewer for summarizing key background information on the topic and for highlighting previous publications, which we also cite in our manuscript. We would like to emphasize, however, that the present study goes significantly beyond our previous report (ref. 39). In that earlier work, we showed that Rad52-Rad51 interaction is not strictly required for Rad51 filament formation, we also observed that Rad52 mutants disrupting this interaction suppress filament toxicity in *srs2Δ* cells and produce filaments that are more sensitive to Srs2-mediated disassembly. These findings led us to propose that Rad52 also stabilizes Rad51 filaments, though the mechanism was unknown.

Here, we provide the first detailed structural description of the Rad52-Rad51 interface, revealing an extended 85-residue Rad52 segment using NMR, AlphaFold, and SAXS. We show that Rad52 binds a single Rad51 monomer in competition with Rad51-Rad51 oligomerization, supporting a model in which Rad52 acts as a chaperone to isolate monomers and load them onto ssDNA.

As the reviewer notes, we tested this model through a comprehensive mutational analysis. Importantly, we extended this approach to assess the role of each contact site not only in mediator activity, but also in Rad51 filament formation by live microscopy and competition with Srs2—an unexplored aspect until now.

1) There are a few specific points which could be improved, for example including western blots to accompany the quantifications of the Rad52-Rad51 co-IP data (e.g., in Supplementary Figure S3G).

We added the western blots in addition to the quantification in Supplementary Figure S3G and 3H of the revised manuscript (cited page 13 of the main text).

Referee response: The WBs are a valuable addition, although one might argue that showing immunodetection for IPed Rad51 rather than WCE Rad51 would be the appropriate control here.

Authors response:

We thank the reviewer for this suggestion. Unfortunately, this control could not be performed due to technical limitations. As noted in the legends of Supplementary Fig. 3G, Fig. 7A, and Supplementary Fig. 7C, as well as in the "Co-immunoprecipitation" section of the Materials and Methods, we were unable to directly detect Rad51 in the immunoprecipitated fractions because it migrates at the same position as the heavy chain of the anti-Rad51 IgG. Consequently, the efficiency of Rad51 immunoprecipitation could not be assessed by direct immunodetection. Nevertheless, the absence of Rad52 in *rad51Δ* immunoprecipitates confirmed that the Rad52-FLAG signals observed were specific to the Rad52-Rad51 interaction.

2) It is also not clear how the authors rationalize that *rad51-T161R* mutant shows complete loss of Rad52 binding

yet only a mild MMS sensitivity phenotype (Fig. 7), whilst structure-guided point mutations in Rad52 that lose interaction with Rad51 resulted in MMS hypersensitivity (see conclusion on page 9, “Interestingly, mutants that exhibited a complete loss of interaction with Rad51, as assessed by co-immunoprecipitation (coIP), were sensitive to low doses of MMS (0.01%), highlighting the functional importance of this interaction.”).

The sentence on page 9 cited by the reviewer refers to a general correlation between loss of Rad52–Rad51 interaction and MMS sensitivity, without yet detailing the variability among mutants. To clarify this point, we have specified page 9 in the revised manuscript that mutants losing Rad52–Rad51 interaction show MMS sensitivity to varying degrees.

As noted in the original text (page 13), “rad52-F316A and F337A (anchors 1 and 2) exhibit higher sensitivity to MMS (Figure 3D) compared to rad52-Y376A (anchor 3).” A similar gradient between mutants was also observed after γ -irradiation (page 13 of the original text). On page 18 we compared rad51-T161R mutant with rad52-Y376A, stating that they display similar MMS sensitivity: “...comparable to that of rad52-Y376A cells (Figure 7B).” To avoid any confusion, we have now explicitly added page 18 a reference to Figure 3D in the revised manuscript to clarify this comparison.

We would like to emphasize that while all three anchor mutations disrupt Rad52–Rad51 interaction (as shown by NMR and co-IP), their *in vivo* effects are more nuanced. As emphasized in the discussion, we stress the importance of combining *in vitro* and *in vivo* approaches to detect subtle phenotypes that may be influenced by additional cellular factors. The range of sensitivities observed supports the distinct functional contributions of each anchor to Rad51 filament dynamics.

Referee response: OK.

3) Generally speaking, the data is of high quality, the protein modelling is appropriately presented and validated, and the conclusions are sound. Having said that, there is a major concern that the structural and functional results in this study largely confirm existing data and the conceptual advance appears to be limited.

In their preceding 2018 study (ref. 39), the authors concentrated on a region within Rad52 spanning amino acid residues 316 to 379, which contains two previously established Rad52-binding motifs (FVTA) at position 316-319 and (YEKF) at position 376-379 (defined by the Sung and Shibata/Kurumizaka groups in refs. 51 and 53).

In the present MS, it is roughly the region demarcated by these known Rad52 motifs that the authors re-label as an 85 amino-acid Rad51 interaction domain (spanning residues 310-395). This domain is found to contain a third motif (residues 337-367) supporting Rad51 binding. An extensive mutational analysis paired with DNA damage-sensitivity spot assays, co-immunoprecipitation of Rad52 and Rad51, and recombination-dependent DNA repair genetic assays is largely consistent with the notion that, as predicted by AlphaFold, there are three motifs (the two previously defined FVTA and YEKF motifs, as well as a related motif starting at F337 of Rad52) mediating three-pronged Rad51 monomer interactions and supporting Rad51 recruitment and nucleoprotein filament formation on single-stranded DNA.

We thank the reviewer for acknowledging the quality of our data.

Contrary to what is stated by the reviewer, our 2018 study (ref. 39) focused solely on a narrow segment of Rad52 (residues 371–387), corresponding to the YEKF motif. In contrast, the present work defines and characterizes, for the first time, the entire Rad51-binding domain of Rad52 (residues 310–395), using a combination of NMR, AlphaFold modelling, SAXS, and extensive *in vivo* functional assays.

The two previously known motifs (FVTA and YEKF) were primarily characterized *in vitro* (ref. 51 and 53), with only limited *in vivo* analysis that did not account for Srs2-dependent effects. In this study, we identify and validate a third functional Rad51-binding motif, centered on F337, and demonstrate that Rad52 uses three distinct “anchors” to engage Rad51 monomers. Notably, we show that anchors 1 and 2 compete with the Rad51–Rad51 interface, thereby directly influencing filament formation, while anchor 3 could play a specific role in protecting filaments from Srs2 disassembly, with less impact on filament assembly itself.

These findings go well beyond confirming previous data. They define a cohesive structural and functional framework for Rad52’s mediator activity, clarify the differential contributions of each interaction site to filament dynamics, and establish a mechanistic link between Rad52 architecture and its dual role in Rad51 loading and filament stabilization.

4) While the key literature is cited, the findings within this MS are not well integrated with the existing literature. For the most part, the results presented are congruent with the current consensus on Rad52’s mediator role. For example, section 1 of Results “A large segment of 85 residues in the center of the fully disordered Rad52 C-terminal domain interacts with Rad51” and Fig 1 overlap largely with findings in refs. 51 and 53. However, refs. 51 and 53 are not cited or discussed in this section, which makes it difficult for the reader to place the MS in its proper context and identify where new insight arises.

Refs. 51 (Krejci et al. 2002) and 53 (Kagawa et al. 2014) are cited in the Introduction, prior to the presentation of our results. To help readers better contextualize our findings in light of these previous studies, we have now added explicit citations to refs. 51 and 53 in line 3 of the first paragraph of the Results section (page 5).

However, we would like to emphasize that these studies did not provide structural data to analyze the Rad52–Rad51 binding mode, as we do in Figures 1 to 4. Furthermore, both refs. 51 and 53 did not consider the contribution of Srs2 activity and incorrectly concluded that Rad52–Rad51 interaction is absolutely required for Rad51 filament formation. In contrast, our results demonstrate that mutants affecting this interaction can still form Rad51 filaments, but these filaments are highly sensitive to Srs2-mediated disassembly. Again, the comprehensive approach presented in this manuscript—including structural characterization, *in vitro* binding assays, and *in vivo* phenotypic analysis—provides a more integrated and robust framework for understanding

Rad52's mediator activity and its role in regulating Rad51 filament dynamics.

Referee response: On re-reading the revised version of the MS, I still feel that previous work (ref. 51, 53) is not appropriately credited with defining the FVTA and YEKF Rad51-binding sites which are confirmed herein. The way the FVTA motif is introduced on p.4 is rather cryptic ("The Rad51 polymerization motif FVTA is conserved in Rad52 C-terminus and essential to disrupt Rad51 oligomers.") and the sentence makes no reference to the literature or the fact that the FVTA site has been defined as a conserved Rad51-binding domain with a notable similarity to BRC domains (ref. 53).

Authors response:

The sentence "*The Rad51 polymerization motif FVTA is conserved in the Rad52 C-terminus and is essential to disrupt Rad51 oligomers*" should be read together with the preceding and following sentences: "*The Rad52 C-terminus has been reported to disrupt Rad51 oligomers and form a heterodimeric complex with Rad51, which may be an important feature of Rad52 mediator activity*⁵³. *The Rad51 polymerization motif FVTA is conserved in the Rad52 C-terminus and is essential for disrupting Rad51 oligomers. A similar domain, FxxA, is found in the BRC repeats of BRCA2 and also disrupts Rad51 oligomers*^{23,54-57}."

In our view, this passage clearly indicates that the FVTA motif is conserved and shares similarity with the FxxA motif in BRCA2 BRC repeats. Reference 53 is cited accordingly.

Similarly, the Sung lab has shown by sequence alignment and mutational analysis that a second Rad51-binding site exists within Rad52 (YEKF) and is conserved (ref. 51). This should be made explicitly clear in the Introduction, which will help the reader appreciate where the data presented in the current MS confirms previous findings and conclusion, and which additional insight is won herein.

Authors response:

We have added a reference to the YEKF motif in the Introduction, as suggested: "*Two conserved Rad51 binding motifs, FVTA and YEKF, have been previously described, using a residue numbering that starts 33 amino acids downstream of the first start codon have been described previously*^{51,52,53}".

Adding refs. 51 and 53 in the first para of Results (p.5; "Although the C-terminal region of Rad52 (...) mediates essential interactions with the recombinase Rad51 [refs. 51,53] (...)") is an important improvement, but I fear the significance of this work will be lost on the reader. I strongly recommend the authors make it clear that these previous papers have defined clearly demarcated motifs that form the downstream and upstream boundaries of the 85-residue region defined in the current MS and that these motifs are equivalent to what is described here as 'anchor 1' and 'anchor 3'. This will provide much better context and, I feel, takes nothing away from the discovery of 'anchor 3' or the structural and functional implications described in the current MS.

Authors response:

We have clarified this point in the description of conserved motifs 1, 2, and 3 in the Results section:

"*When restricted to fungi, the alignment reveals three distinct motifs (Figure 1F, top panel): motif 1 (from residue 312 to 322, which contains the previously described FVTA motif⁵³), motif 2 (from residue 337 to 367), and motif 3 (from residue 371 to 392, which contains the previously described YEKF motif⁵¹)*".

The potential impression that specific anchor points (1 and 3) have not been previously defined at amino acid level should be avoided, in particular since historic Rad52 residue numbering in refs. 51 and 53 could make it difficult for some readers to spot the congruence between the earlier work and the current paper. Appropriate textual changes are required before publication.

Authors response:

To address the reviewer's concern about potential confusion due to residue numbering, we note that this point is already clarified at the beginning of the Results section: "*Note that the Rad52 amino acids are numbered from the first AUG codon in RAD52 mRNA*⁵⁹".

This clarification is also present in the Materials and Methods section.

In addition, as already mentioned, we have clarified in the Introduction that the FVTA and YEKF motifs had been previously identified as conserved Rad51-binding sites. We also specify that these earlier studies used a residue numbering starting 33 amino acids downstream of the first start codon.

We hope these revisions fully address the reviewer's concern and provide the appropriate context and credit to previous foundational work.

5) A similar criticism applies to the subsequent section and Fig. 2, which also appears largely redundant with the literature (Rad52 binding to Rad51 promoting Rad51 oligomer dissociation, e.g. ref. 53).

While Rad52's role in promoting Rad51 oligomer dissociation has been previously shown (e.g., ref. 53), the data presented in Figure 2 provide additional structural insight through SAXS analysis. This approach allows us to determine the stoichiometry of the complex and to compare it across various mutants under identical experimental conditions. We have now added page 7, a reference to the relevant literature in this section of the

manuscript, as follows: "These results are fully consistent with previous analyses of this complex by SEC (ref. 53)."

Referee response: OK.

6) With respect to the main conclusion and working model in Fig. 8, similar models have been published previously for BRCA2 as well as Rad52, and while some of these previous publications are cited, this is not clearly discussed in the Discussion section to identify new advances specific to the current MS. Regarding the similarities and differences between BRCA2 and Rad52, the revised manuscript now includes a more complete, dedicated paragraph at the end of the Discussion section, in response to Reviewer #1's comment. This paragraph discusses their respective roles in Rad51 filament stability and highlights that anchor 3, which contributes specifically to filament protection, is absent from BRCA2. This underscores the unique ability of Rad52 to bind Rad51 through an extended interaction segment comprising three distinct regions, each exerting a subtle but specific impact on Rad52's mediator activity.

Referee response: OK.

7) A discussion of the model for Rad52-Rad51 interactions/Rad51-chaperoning presented recently by Deveryshetty and co-workers (ref. 25) is missing. The Deveryshetty model (Fig. 8 in ref. 25) is very similar to the one presented herein (Fig. 8 in the current MS). Moreover, the authors do not relate the ideas illustrated in their model (Fig. 8) to the suggestions of Deveryshetty and co-workers who, besides C-terminal Rad52-Rad51, also show N-terminal Rad52-Rad51 interactions that are important for Rad52's mediator function. We agree that a comparative discussion of our model with the one proposed by Deveryshetty et al. (ref. 25) is a valuable addition to the manuscript. We have therefore added, a sentence in the Discussion page 21 section that refers directly to this study: "This oligomerization ability is conferred by its N-terminal annealase domain, whose structure was recently solved (ref. 25) (Supplementary Figure 1A). In that study, the authors proposed that the ssDNA-binding property of this domain, together with the association of Rad51 with the Rad52 C-terminal domain, could facilitate the transfer of Rad51 onto single-stranded DNA." Ref. 25 is also now cited at the end of the Discussion about the multisite organization of Rad52 and BRCA2 mediators. Also we gave more details about the model in the legend of Figure 8. Importantly, our model differs substantially from that of Deveryshetty et al., both in focus and in the nature of the supporting data. Their study primarily addresses the structural characterization of the N-terminal annealase domain and includes the C-terminal region in their model, but without experimental validation. In a complementary manner, our study focuses on the C-terminal domain of Rad52, and we integrate relevant insights from ref. 25 into our model. However, we do not attempt to depict interactions between the annealase domain and DNA, as this falls outside the scope of our experimental data.

Referee response: OK.

8) Kagawa (2013) and co-workers (ref. 53) have previously noted the striking similarity between the Rad51-binding motifs in Rad52 and the RAD51-binding BRC motifs in BRCA2, and they have published a homology model for the role of the Rad52 and BRCA2 motifs in sequestering monomeric Rad51/RAD51 to mediate proper loading onto single-stranded DNA/nucleoprotein-filament nucleation and growth (see Fig. 7 in ref. 53, similar to the iteration in Supplementary Fig. 4 in the current MS). Indeed, the very interesting article published by Kagawa and co-workers (ref. 53) identified a clear similarity between the FxxA motifs of Rad52 and BRCA2. Figure 7 of that study presents the structure of Rad51 in complex with BRC4 by Pellegrini et al (ref 14 in our manuscript), and a sequence alignment of Rad52 homologues and, based on these information, this paper proposes that the Y376EKF motif in Rad52 could bind Rad51 similarly to F1524 in the BRC4 motif of BRCA2. However, this homology-based model is incorrect. Our structural analysis demonstrates that Y376 binds to a completely different site on Rad51 compared to F1524. To clarify this point, we have added Supplementary Figure 8, which directly compares the binding modes. Our study thus resolves a key inaccurate hypothesis from ref. 53, which arose due to the absence of reliable structural data in that earlier work.

Referee response: The conceptual point made by Kagawa et al. should be acknowledged in this context, with citation of ref. 53, regardless of whether their working hypothesis in the 2013 paper faithfully reflects the detailed protein interaction or not. I can't see where this is acknowledged, but if not done already, the insightful prediction by Kagawa et al. should be mentioned.

Authors response:

We appreciate the reviewer's suggestion. While we acknowledge that the study by Kagawa et al. (ref. 53) raised an interesting hypothesis, we believe that their proposed model does not accurately reflect the current understanding of the Rad52-Rad51 interaction based on available experimental evidence. For this reason, we have opted not to include this reference in this part of the Discussion section, in order to avoid potential confusion. We hope the reviewer understands our position on this matter.

9) Based on their structural data, Deveryshetty and co-workers (ref. 25) envisage a Rad51 chaperone model, where the C-terminal tails of Rad52 extending from the Rad52 homodecameric ring (formed by the Rad52 N-

terminal domains) sequester Rad51 molecules and deliver them for assembly onto DNA (ref. 25). The model proposed by Deveryshetty et al. (ref. 25) suggests that each C-terminal tail of Rad52 binds two Rad51 molecules—an assumption not supported by experimental data. Furthermore, their study does not propose a chaperone model per se, but rather a scaffolding model in which each C-terminal region recruits multiple Rad51 molecules to generate a sufficient local concentration for filament nucleation. Notably, the concept of a Rad51 chaperone is never explicitly mentioned in the text of their paper.

Referee response: Semantics aside, the Deveryshetty model is conceptually very similar to the one presented here, and this is now mentioned in the revised MS - response point 7.

10) An interesting point of novelty within the current MS is the three-pronged (rather than two-pronged) nature of interaction between Rad52 and Rad51. In addition, by structural analysis/modelling, the authors succeed in providing an improved view of Rad52-Rad51 interactions. The work is solid, interesting, and should be published; however, it does not provide the kind of conceptual advance over what we know about Rad52/BRCA2-Rad51/RAD51 mediator function that would make the MS a strong candidate for publication in Nature Communications.

As already mentioned, our study not only provides robust structural data, as acknowledged by Reviewer #2, but also includes phenotypic analyses of three Rad52 mutants that subtly affect its mediator activity. Taken together, these findings support a new conceptual framework for Rad52 function: the idea that Rad52 acts as an assembly chaperone for Rad51, both promoting filament formation and protecting it from Srs2-mediated disassembly. This dual role—mechanistically defined and experimentally supported—represents a meaningful conceptual advance beyond current models of mediator function for Rad52 and BRCA2.

Reviewer #3 (Remarks to the Author):

The comments and concerns I raised in the previous round of peer review have been addressed to my satisfaction. I would recommend this paper be accepted for publication.

Authors response:

We thank the reviewer for their help in improving the manuscript.